



# A novel aerosol filter sampler for measuring the vertical distribution of ice-nucleating particles via fixed-wing uncrewed aerial vehicles

Alexander Böhmländer[1], Larissa Lacher[1], David Brus[2], Konstantinos-Matthaios Doulgeris[2], Zoé Brasseur[3], Matthew Boyer[3], Joel Kuula[2], Thomas Leisner[1], and Ottmar Möhler[1]

[1]Institute of Meteorology and Climate Research, Karlsruhe Institute of Technology, Karlsruhe, Germany
[2]Atmospheric Composition Research, Finnish Meteorological Institute, Helsinki, Finland
[3]Institute for Atmospheric and Earth System Research/Physics, Faculty of Science, Unisversity of Helsinki, Helsinki, Finland

**Correspondence:** Ottmar Möhler (ottmar.moehler@kit.edu)

**Abstract.** A mobile sampler for the collection of aerosol particles on an uncrewed aerial vehicle (UAV) was developed and deployed during three consecutive Pallas Cloud Experiment campaigns in the vicinity of the Sammaltunturi Global Atmosphere Watch site (67°58' N, 24°7' E, 565 m above sea level). The sampler is designed to collect aerosol particles onto Nuclepore filters, which are subsequently analysed for the temperature-dependent number concentration of ice-nucleating particles of the sampled aerosol with the Ice Nucleation Spectrometer of the Karlsruhe Institute of Technology (INSEKT). This setup is an easy and flexible way to connect INP concentration measurements with cloud microphysics. The sampler was flown with a fixed-wing UAV in different altitudes up to 1000 m above ground level. The total flight time ranges from 1 hour to more than 1.5 hours, depending on environmental conditions. Pressure, temperature and relative humidity are also measured to provide information about the meteorological flight conditions. The flow over the filter was maintained by a micro-diaphragm pump, providing around 10 standard litres per minute over a small filter (diameter of 25 mm) and around 11 standard litres per minute over a larger filter (diameter of 47 mm) at a pressure corresponding to 500 m above sea level. For a typical flight time of 1.5 hours, this results in a sampled air volume of about 930 to 1000 standard litres per flight, giving an INP detection limit of approximately $1.1 \times 10^{-3}$ and $1.0 \times 10^{-3}$ INPs per standard litre, respectively. For comparison to the flight results, a similar setup was deployed at ground level. The comparison shows a clear distinction from the water and handling blank background for both setups, proving the technical feasibility of the setups. Furthermore, for some flights, a shift between the two INP populations can be seen, indicating that ground-based INP measurements deviate from the samples collected on-board the UAV.

## 1 Introduction

Ice-nucleating particles (INPs) are a rare subset of aerosol particles, which induce primary ice formation inside clouds, and therefore play special and important roles in aerosol-cloud interactions. The formulation and quantification of these interactions are largely uncertain in current weather and climate models (Boucher et al., 2013; IPCC, 2021; Murray et al., 2021). While cloud water droplets can freeze homogeneously only below about -35 °C (e.g., Pruppacher and Klett, 1997), INPs decrease the threshold for ice nucleation, and therefore enable water droplets to freeze well above -35 °C (e.g., Vali, 1996; Pruppacher



and Klett, 1997; Murray et al., 2012; Hoose and Möhler, 2012; Kanji et al., 2017). In this way, INPs significantly contribute
to primary ice formation, which impacts the depletion of supercooled water inside mixed-phase clouds (e.g., Wegener, 1911;
Bergeron, 1935; Findeisen, 1938; Shi and Liu, 2019). The ratio of ice crystals and supercooled water droplets also largely
impacts the cloud albedo and therefore the radiation budget of Earth (e.g., Korolev et al., 2017; Lohmann, 2017; Storelvmo,
2017; Desai et al., 2019; Shi and Liu, 2019). Furthermore, about 50 % of all precipitation events of more than 1 mm per day
are linked to the occurrence of the ice phase in the cloud, and this value increases to more than 90 % for polar regions (Field
and Heymsfield, 2015; Mülmenstädt et al., 2015; Heymsfield et al., 2020). Most field observations measure INPs at ground-
or aircraft-level (e.g., DeMott et al., 2010, 2017; Kanji et al., 2017; Schneider et al., 2020; He et al., 2021), and it is not yet
sufficiently understood how to connect ground-based INP measurements with cloud formation processes. While aircraft can
be used to measure INP concentrations at the level of cloud formation, these measurements are not feasible for longer-term
studies due to the high operational costs.

Recently, uncrewed aerial vehicles (UAVs) have become a focus for atmospheric measurements (e.g., Bärfuss et al., 2018;
Lampert et al., 2020; Marinou et al., 2019; Villa et al., 2016; Yu et al., 2017). Some studies have been performed to measure
INPs on a UAV (Schrod et al., 2017; Bieber et al., 2020) or with balloon-based sampling systems (Porter et al., 2020). Bieber
et al. (2020) used a multicopter to measure biogenic INPs up to 100 m above ground level (agl) with flight times of 10 minutes.
Longer sampling times (< 90 minutes) and higher altitudes (< 2.5 km agl) were reached with a setup using two fixed-wing UAVs
(Schrod et al., 2017), enabling measurements of the vertical INP distribution during dust events in the Eastern Mediterranean. A
balloon-based size-resolved INP sampler was developed by Porter et al. (2020) and was deployed during campaigns in Hyytiälä
(southern Finland), Leeds (northern England), Longyearbyen (Svalbard, Norway), and Cardington (southern England). The
payload is tethered at a specific height (< 2.3 km) with a winch and can sample up to 11 hours. In general, the sampling time
as well as the sample flow over a filter determine the lower detection limit for INPs. This lower detection limit is especially
relevant at higher subzero temperatures, where the INP concentration is orders of magnitudes lower than at lower temperatures
(e.g., DeMott et al., 2010; Kanji et al., 2017).

In this study, we present a filter-based aerosol sampler flown on a fixed-wing UAV. A fixed-wing UAV is able to provide
longer flight duations as well as a constant airspeed compared to multicopter UAVs. The main advantage of a UAV is the
flexibility of use as well as the low operational costs compared to an aircraft or balloon-borne setup. No runway is needed,
and the fixed-wing can be started and deployed by two people in a matter of minutes, only hindered by flight regulations and
weather restrictions.

## 2 Experimental

### 2.1 Flight platform

The UAV used to carry the payload is a Skywalker 1830 model year 2015 (customised by Yugen Oy). The Skywalker is a
fixed-wing UAV with a wingspan of 1830 mm and a maximum takeoff mass of 3000 g. The maximum payload weight of
the Skywalker is approximately 1200 g. The fuselage is 220 mm long with a width of 120 mm and can be accessed via two





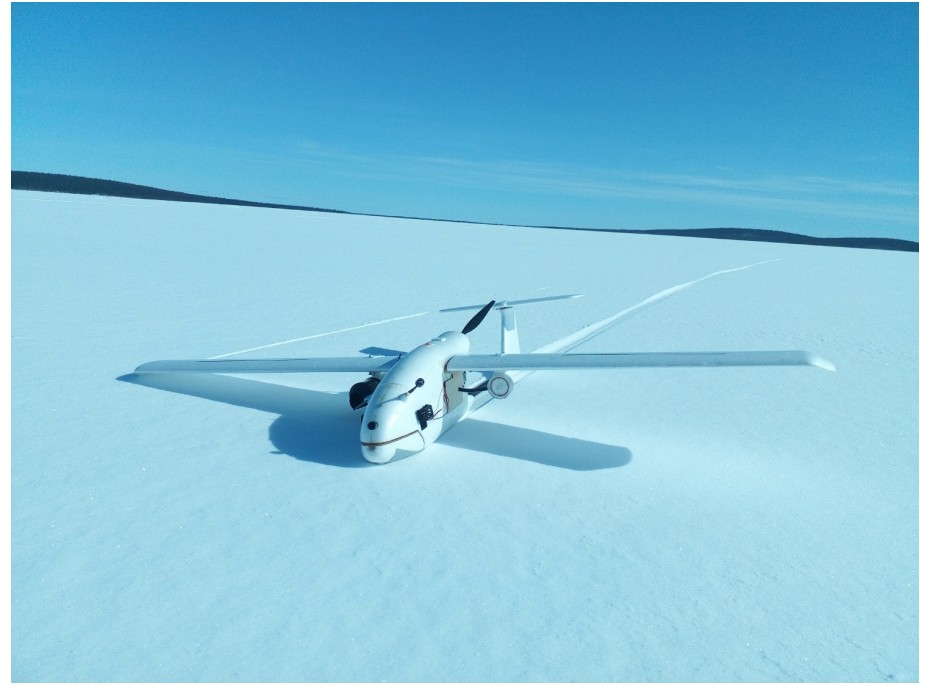

**Figure 1.** The Skywalker shown with the final setup after a successful flight. The filter holder is installed with a tube connection through the wooden side plate. The inlet is blocked until launch. On the right, the ambient sensor (BME280, Bosch) can be seen, and the airspeed sensor is visible on the left side (ASPD.7002, Matek Systems).

openings covered with wooden plates (150 mm x 75 mm) on both sides. The Skywalker is powered by two LiIon batteries (4S, 7000 mA h) connected in parallel, providing a maximum flight time of about 2 hours in ideal weather conditions. It is a glider-type airframe, enabling long flight duration at one altitude at low power consumption. It can be launched by hand and just needs a flat surface (i.e. grass) for landing on the belly. These features make the Skywalker a well-suited UAV for filter-based aerosol sampling. The Skywalker contains additional components, such as a flight controller using arduplane firmware 4.06 (F405-WING, Matek Systems), an analogue airspeed sensor (ASPD-7002, Matek Systems) and a compass module (M8Q-4883, Matek Systems). The data from the compass module is used to track the UAV flight path via global navigation satellite system (GNSS) data.

## 2.2 Payload

The aerosol sampling setup developed here was tested and improved during three field campaigns in Pallas, Finland. During the first campaign in autumn 2020, the whole payload was located inside the fuselage and the aerosol inlet was located below the left wing, resulting in a sampling line with two 90° bends. This setup proved the technical feasibility of measuring INPs with sample times between 45 and 90 minutes in regions with low aerosol concentrations, such as northern Finland. For the second version deployed in spring 2021, the filter holder was placed below the batteries in the front of the UAV, with a short





**Table 1. UAV-based aerosol sampler.** Description of the used components in the UAV-based aerosol sampler. The total weight of the sampling unit is about 870 g. Sensor specifications are from the respective data sheets (Sensirion, 2013; Bosch, 2020; Sensirion, 2021).

| Name | Description | Details |
|---|---|---|
| Inlet[a] | Inner diameter: 6 mm | stainless steel, antistatic tubing |
| | Length: 17.2 cm (14.2 cm[b]) | |
| Filter holder | Diameter: 25 (47[c]) mm | polypropylene, Whatman, 420200 (420400[c]) |
| | Weight: 10 (62[c]) g | |
| Filter | Diameter: 25 (47[c]) mm | Nuclepore, Whatman, 110637 (111137[c]) |
| | Pore size: 0.4 μm | |
| Flow meter | Pressure drop: $< 25 \, \mathrm{hPa}$ | Sensirion SFM4100 Air |
| | Flow range: $0–20 \, \mathrm{l_{std} \, min^{-1}}$ | |
| | Accuracy: 0.15 % of full scale or 3 % of reading, whichever is bigger | |
| Pump | Weight: 380 g | KNF, NMP850.1.2KPDC-B HP |
| | Flow at 1013 hPa: $15 \, \mathrm{l \, min^{-1}}$ | |
| | $I_{\mathrm{max}} = 2.4 \, \mathrm{A}$, $U = 12 \, \mathrm{V}$ | |
| Single board computer | Weight: 9 g | Raspberry Pi Zero WH |
| LiIon battery | Weight: 280 g | 4S |
| | Capacity: 3300 mA h | |
| Ambient sensor 1 | $T$: -40–85 °C | Bosch Sensortec BME280 |
| | RH: 0–100 %RH | |
| | $p$: 300–1100 hPa | |
| Ambient sensor 2 | $T$: -40–125 °C | Sensirion SHT40 |
| | RH: 0–100 %RH | |

[a]The third version of the setup does not use an inlet.

[b]The second version has a decreased length of the inlet.

[c]The third version of the setup contains the larger filter holder.

straight horizontal tube as the aerosol inlet upstream of the filter. This change enhanced the sampling efficiency of larger aerosol particles due to fewer bends (see app. B for an estimation of the transport efficiency for the different setups). For the third setup used during autumn 2021, the filter holder was placed outside of the fuselage, below the wing (setup depicted in Fig. 1). In the following, this third and final setup is described in more detail (see also Table 1). In our final setup, the payload contains a micro-diaphragm pump (NMP850.1.2KPDC-B HP, KNF), which provides a flow of $15 \, \mathrm{l \, min^{-1}}$ at standard conditions (KNF). The pump weighs 380 g and draws a maximum current of 2.4 A at a voltage of 12 V. The pump is connected to a mass flow meter (SFM4100, Sensirion) to monitor the flow during the sampling, which depends on the ambient pressure conditions, and therefore varies at different sampling altitudes. The mass flow meter is read out with a single board computer (SBC; Raspberry





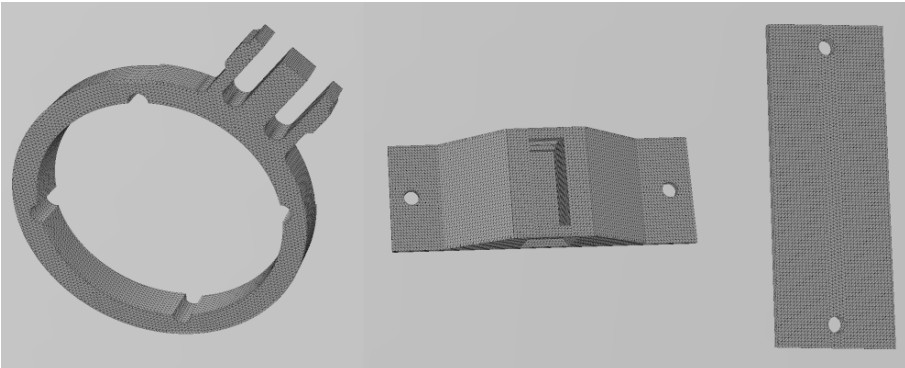

**Figure 2.** 3D printed parts for the filter holder mounting to the plane. From left to right: filter holder mount, mount base, mount backplate. The mount base is connected to the wing by the backplate, while the filter holder snaps into the base and secures the position of the filter holder.

Pi Zero WH, Raspberry Pi) at a frequency of 1 Hz, and is used to identify issues during the operation, i.e. connection failures or
clocking of the filter pores. Upstream of the flow meter, a plastic filter holder (420400, Whatman) with a diameter of 47 mm and a weight of roughly 62 g is connected and mounted below the right wing (see Fig. 1). The mount is a 3D printed piece that can be quickly connected and disconnected to the wing (see Fig. 2). The whole payload is powered by a LiIon battery (3300 mA h) for more than 2 hours. In addition, the SBC is also used to read out two sensors at the front of the UAV (BME280, Bosch Sensortec, and SHT40, Sensirion), providing temperature $T$, pressure $p$, and relative humidity RH data with an uncertainty
of ± 1.0 K (0.2 K for SHT40), ± 1.0 hPa (only BME280) and ± 3 %RH (1.8 %RH for SHT40), respectively (Bosch, 2020; Sensirion, 2021). The pressure data of the BME280 is used to calculate the flow during the flight (see app. C). All components are detailed in Table 1, and a schematic view is presented in Figure 3. A second identical ground-based sampling system consists of the same components as the UAV sampler.

### 2.3  Typical flight operation

Prior to each flight, the filter holder is loaded with a Nuclepore filter (111137, Whatman). While the filter holder is installed, the inlet of the sample tube is closed with a cap, which is removed right before the start of the flight. During the preparations for each flight, the autopilot mission is uploaded onto the flight controller using MissionPlanner (version 1.3.74), the SBC is connected to the batteries, and the respective scripts are started to read out the flow meter and the meteorological sensors. After the UAV is hand-launched, it is flown manually to the designated altitude. Once the UAV reaches the targeted altitude,
the autopilot is turned on to initiate a loiter command that steers the UAV in circles with 200 m radius above the measurement field, keeping the same altitude during the remainder of the flight (see Fig. 4). At the same time, the pump is turned on remotely via radio switch to start sampling aerosol particles, making sure that aerosol particles are only actively sampled at one altitude. The start-up procedure typically takes less than 10 minutes, depending on the targeted sampling altitude. Once the pump of the UAV sampler is turned on, the ground-based aerosol sampling (location shown in Fig. 4, red cross) is started





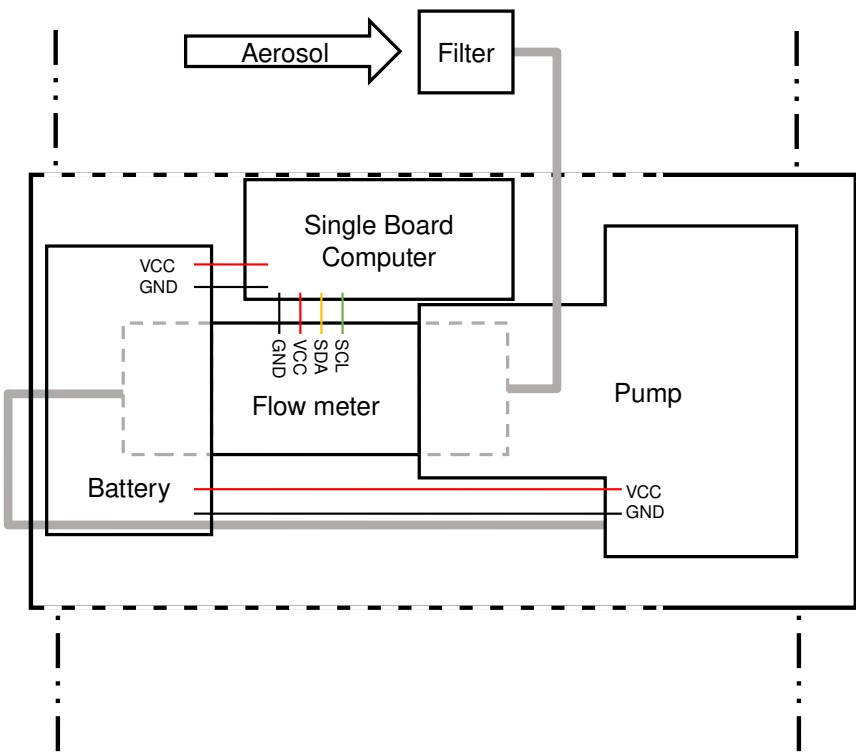

**Figure 3.** Schematic view from the top of the setup. The fuselage is shown with the wings in the bottom and top marked by the vertical dash-dotted lines. The cover plates are indicated by the dashed horizontal lines. The pump connections are shown for the filter in grey with an arrow to indicate the flow of the aerosol. The electrical connections are drawn for the power (VCC, voltage at the common collector; GND, ground) as well as for the data connection via I2C for the flow meter (SCL, serial clock; SDA, serial data). For simplicity, the data connections to the two ambient sensors SHT40 and BME280, which are also read by the single-board computer via I2C, are not shown.

as well, providing a temporal overlap of the collection times for comparing the INP concentrations measured at ground level and UAV flight altitude. Figure 5 shows a typical timeseries of the sensor data during one flight experiment. The SHT40 has a lower uncertainty compared to the BME280 but only measures the temperature and the relative humidity. The pressure data is important for the setup since it is used to calculate the sampling flow.

## 2.4    Filter handling and subsequent offline INP analysis

The Nuclepore filters used for aerosol collection are pre-cleaned with $10\,\%$ $H_2O_2$ and afterwards rinsed with Nanopure water (generated by Barnstead GenPure Pro UV), which was passed through a $0.1\,\mu m$ syringe filter (6784-2501, Whatman). The clean filters are then dried on aluminium foil under a constant clean air flow and afterwards packaged in pairs inside pre-heated aluminium foil. During handling of the filters, forceps that are pre-cleaned the same way and packaged in aluminium foil are used. After aerosol collection on the UAV or with the ground-based setup, the filters are stored in sterile Petri dishes, packed





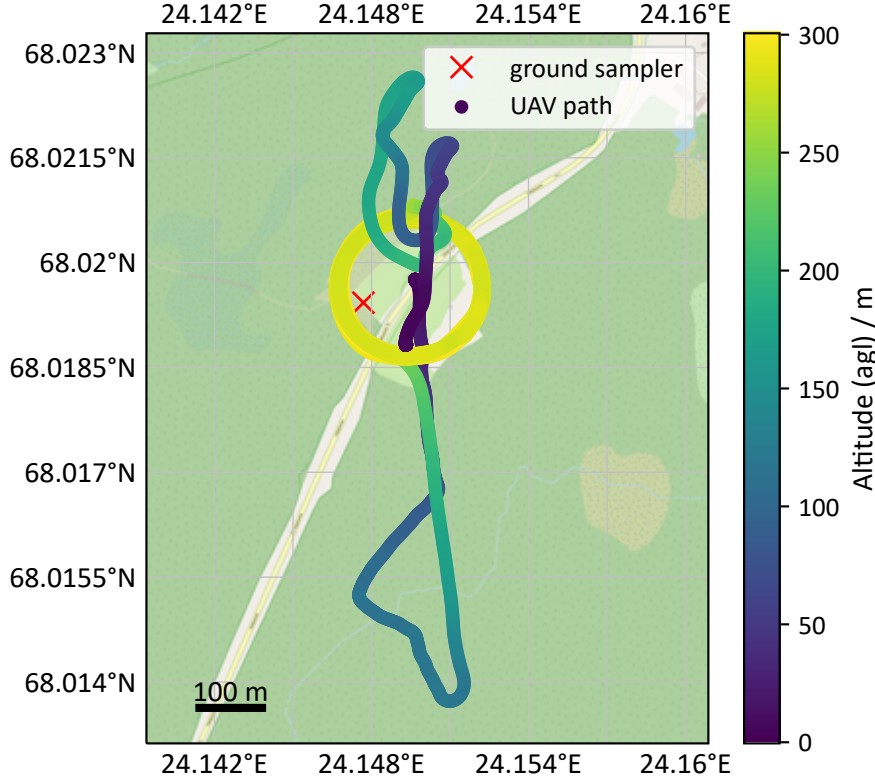

**Figure 4.** Typical flight path of the Skywalker during an experiment at 300 m agl at the sampling location in Pallas, Finland. The colour indicates the altitude agl as given by the GNSS measurements from the flight controller, while a red cross marks the position of the ground sampler on top of a wooden hut (about 2 m agl). Map data from ©OpenStreetMap. Distributed under the Open Data Commons Open Database License (ODbL) v1.0.

inside aluminium foil, and stored until analysed by the Ice Nucleation Spectrometer of the Karlsruhe Institute of Technology (INSEKT, see e.g. Schneider et al. (2020)). The INP background from the sampling method is quantified by taking handling blanks that show possible contaminations during handling. The handling blanks are loaded onto the filter holder; the filter holder is mounted on the UAV, but the pump is not turned on; afterwards, the handling blanks are compared to the sampled filters to make sure that the collected aerosol stems from the measurement and not from contaminations during the handling.

For a more detailed description of potential contaminations and procedures during filter handling, see e.g. Barry et al. (2021).

Before analysis with INSEKT, collected aerosol particles are washed off the filter inside centrifuge tubes with 5–8 ml Nanopure water generated by Barnstead GenPure Pro UV and filtered through an additional 0.1 µm syringe filter (6784-2501, Whatman). After tumbling at 60 rpm (1 Hz) for 20 minutes, the washing water is filled into 64 PCR wells, while 32 PCR wells are filled with the filtered Nanopure water.



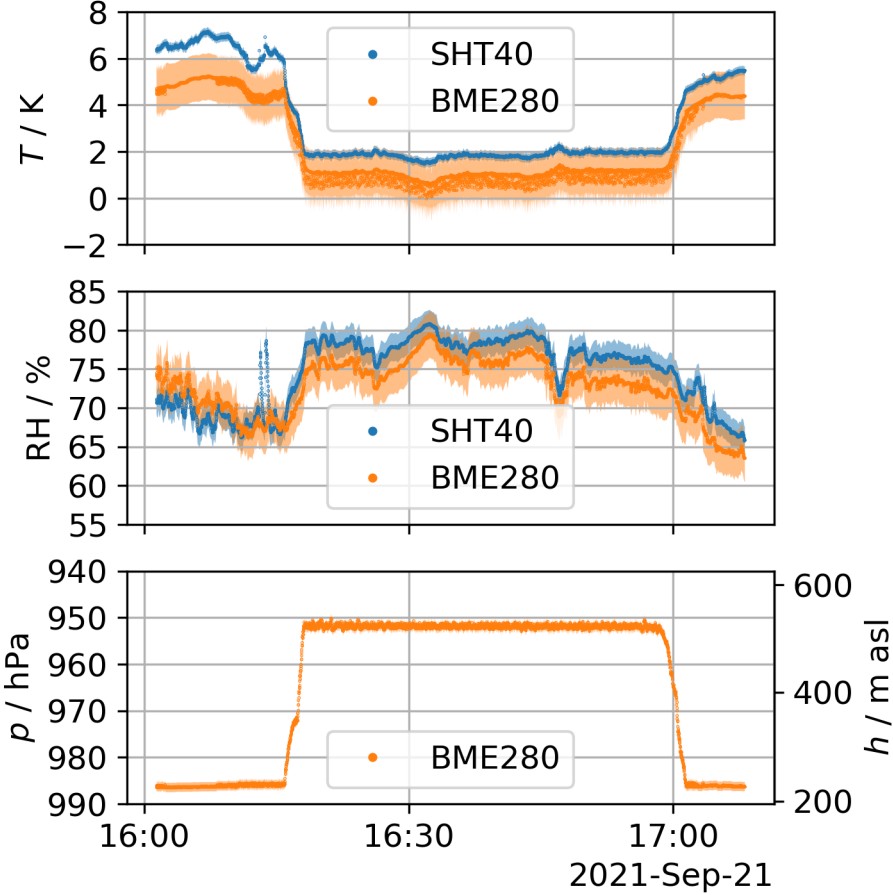

**Figure 5.** Typical sensor data during an experiment at 300 m agl. The three parameters temperature $T$, relative humidity RH and pressure $p$ are shown for the two sensors (blue line, SHT40, orange dotted, BME280) in panels (a), (b) and (c), respectively. In addition, the height is calculated with the barometric height formula and plotted on the right y axis in panel (c), depicting the ascend and descend, as well as the loitering above the ground. The uncertainty bands for both sensors show that the measurements are in agreement. Only the temperature shows a statistically significant difference before and after the flight. This might be due to the fact that the sensors are located on different sides of the UAV, influenced by sun radiation. This difference is decreasing during loitering, where both sensors will be facing the sun at different times during the flight.

INSEKT consists of two aluminium incubation blocks for holding 96-well polymerase chain reaction (PCR) plates (Cat. No. 781368, Brand). The blocks are temperature controlled by a cooling liquid from a cryostat (Proline RP855 for INSEKT1, Pro RP 245 E for INSEKT2, Lauda). The whole setup is enclosed inside a polyvinyl chloride (PVC) box, which is insulated by 2 cm thick ArmaFlex insulation material. The PVC box is topped by a removable anti-reflection coated glass pane. The glass pane protects the samples from contamination with ambient aerosol particles during the analysis. To prevent condensation on

the glass pane, a flow of cooled, dry, synthetic air passes over it. Eight Pt100 temperature sensors (PT100 A 20/050 (NB),





class A, Electronic Sensor GmbH) are placed inside evenly spaced drilled holes inside the aluminium blocks, and their data is read out with a custom-made LabVIEW programme. The Pt100 sensors are additionally calibrated, resulting in a systematic standard deviation of about $0.02\,\mathrm{K}$, while the statistical standard deviation is about one order of magnitude higher. A camera is located above the freezing array, filming the PCR plates through a polarisation filter. The freezing array is cooled down at a rate

of $0.33\,\mathrm{K\,min^{-1}}$. The freezing of a well results in an abrupt change in its recorded grayscale value. From the amount of frozen wells in comparison to the total amount of wells, the liquid fraction, $f_\mathrm{l}$, can be calculated, and from that the INP concentration in the solution, $c_\mathrm{INP}^\mathrm{sol}$, according to

$$c_\mathrm{INP}^\mathrm{sol} = -\frac{d}{V_\mathrm{well}}\ln(f_\mathrm{l})\,, \tag{1}$$

where $d$ is the dilution scale and $V_\mathrm{well}$ the volume of one PCR well ($50\,\mu\mathrm{l}$). The water background, which is obtained by adding

pure Nanopure water to some wells, gets subtracted from the INP concentration in the solution according to its liquid fraction.

Combined with the mass flow over the filter, $F_\mathrm{std}$, and the sampling time, $t_\mathrm{sample}$, the INP concentration in standard litres of air, $c_\mathrm{INP}^\mathrm{air}$, can be calculated via

$$c_\mathrm{INP}^\mathrm{air} = \frac{V_\mathrm{sol}}{F_\mathrm{std}\,t_\mathrm{sample}}c_\mathrm{INP}^\mathrm{sol}\,. \tag{2}$$

For a more detailed look into INSEKT and the used formulas, see Hill et al. (2016); Schneider et al. (2020); Vali (1971).

## 140  2.5  Uncertainty and lower detection limit

The measured INP concentration per standard litre of air has a statistical uncertainty, described by the Wilson interval (Agresti and Coull, 1998). The systematic uncertainty is calculated from the uncertainties of the aerosol sample measurements and the water volumes filled into the PCR wells for the INSEKT analysis, the latter of which results from the pipettes used to fill the PCR wells (Schneider et al., 2020). The systematic uncertainty is roughly two orders of magnitude smaller than the statistical

uncertainty, therefore it is not shown in the plots (see app. A for the detailed uncertainty calculation).

The lower detection limit can be estimated by the condition that a single INP has to exist to initiate freezing, i.e. for a sampled volume of $500\,\mathrm{l_{std}}$, the rough estimate for the lower detection limit for the INP concentration is $c_\mathrm{INP,low}^* = 2\cdot10^{-3}\,\mathrm{l_{std}}^{-1}$. This detection limit also depends on the analysis, since the whole suspension is not used for one analysis. Therefore, a better estimate of the lower detection limit is given by the product of the analysed water fraction and the earlier estimate

$$c_\mathrm{INP,low} = c_\mathrm{INP,low}^*\frac{V_\mathrm{sol}}{V_\mathrm{well}\,n_\mathrm{filled}}\,, \tag{3}$$

where $n_\mathrm{filled}$ is the number of wells filled with the suspension. The lower detection limit is especially important for higher nucleation temperatures, where INPs are generally more rare (e.g., Kanji et al., 2017).

## 3  First application in field campaigns

The newly developed UAV-based aerosol sampler was used and further developed during three Pallas Cloud Experiment cam-

155 paigns, close to the Sammaltunturi Global Atmosphere Watch (GAW) site ($67°58'$N $24°7'$E, $565\,\mathrm{m}$ above sea level (asl),



**Table 2. Description of the campaigns.** During the first campaign, the first setup was used, while the second campaign already featured the improved setup with a straight inlet coming from the front of the UAV. The last campaign features the newest developments with a filter holder connected directly to a wing.

|  | Number of flights | Min / max height agl | Start | End | Setup change |
|---|---|---|---|---|---|
| Campaign 1 | 13 | 100 m / 800 m | 2020-09-22 | 2020-09-30 | proof of concept of the setup |
| Campaign 2 | 12 | 250 m / 1000 m | 2021-04-19 | 2021-04-22 | removal of bends in the sampling line leads to a decrease in sampling losses |
| Campaign 3 | 03 | 150 m / 300 m | 2021-09-20 | 2021-09-23 | shorter sampling line decreases diffusional losses, filter with a bigger diameter provides a lower pressure drop, leading to an increase in flow |

northern Finland, Lohila et al. (2015)), which took place during autumn 2020, spring 2021, and autumn 2021. The measurement site is located in a clean subarctic environment around 180 km north of the Arctic circle. Snow is abundant between November and May, and low-level clouds have a typical occurence of around 40 % during autumn (Hatakka et al., 2003).

A summary of the campaigns is shown in Table 2. A total of 28 flights have been conducted at heights ranging from 100 m agl to 1000 m agl. The flights were conducted at the Finnish Meteorological Institute (FMI) Arctic UAV base, located within a temporary danger area (TEMPO-D Pallas), 7 x 7 km with a ceiling of 2000 m agl and centred around the Sammaltunturi GAW site. This danger area allows the use of uncrewed aircraft beyond visual line of sight. Campaign 1, during autumn 2020, was used to test the first version of the setup in the field (see Sect. 3.1). A second improved version of the setup was tested in campaign 2 during spring 2021 and is described in section 3.2. The final setup was tested between 20[th] and 23[rd] September 2021 (see Sect. 3.3).

## 3.1 Campaign 1

Campaign 1 demonstrated the technical feasibility of the new UAV aerosol sampler in combination with the INSEKT INP analysis. To demonstrate the scientific feasibility, the inverse of the liquid fraction, the frozen fraction, of the UAV and ground filters are compared to their respective handling blank filters taken during campaign 2. Figure 6 shows the frozen fraction as a function of the freezing temperature for all UAV filters, one blank filter, and its respective water background on the left panel. The right panel shows the same for the ground-based filters.

The UAV filter suspensions contain aerosols sampled between 250 and 1000 m agl, whereas the blank filter was handled as described in subsection 2.4. While the blank filter suspension is close to the water background, the UAV filter suspensions show a clear separation from the water background and the handling blank background for temperatures below 253 K. The frozen





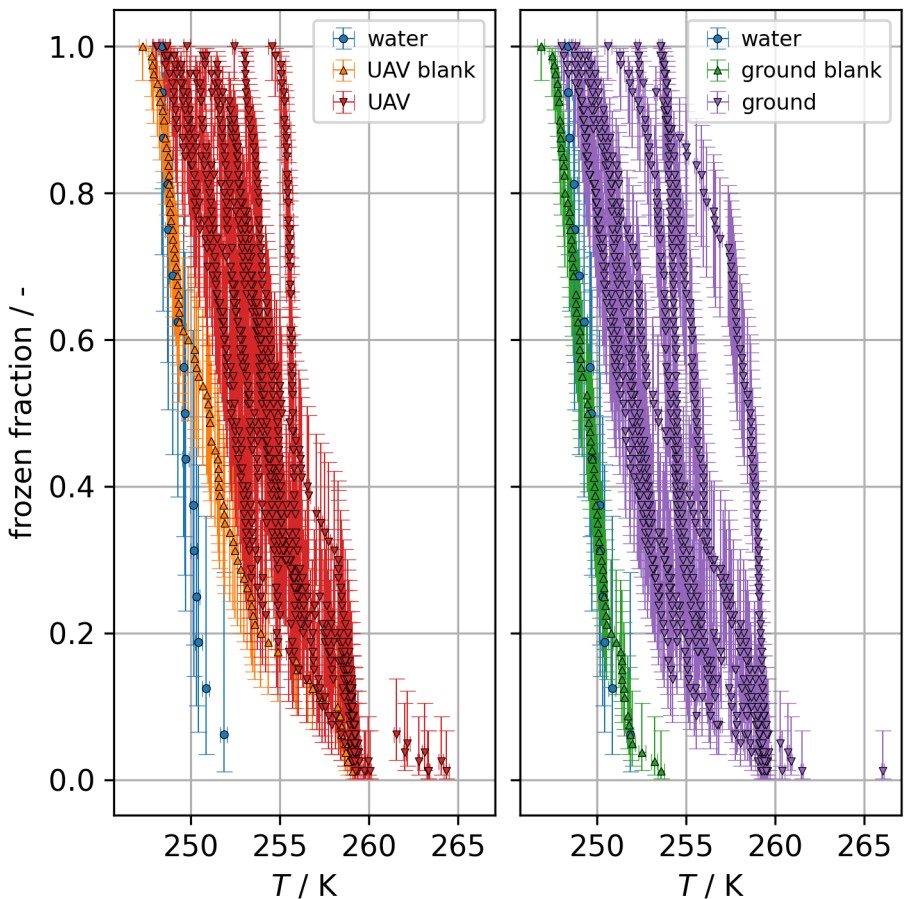

**Figure 6.** The frozen fraction as a function of the freezing temperature $T$ is shown for the UAV filter suspensions from campaign 2 in comparison to its blank filter suspension and the Nanopure water background (left panel). The right panel shows the equivalent for the ground filters. The blanks were handled the same way as the filter, but the pump was not turned on, and the UAV was not flying (see Sect. 2.4).

fraction of the UAV filter suspension starts to freeze at temperatures about 2–7 K higher than the handling blank and the water background, showing that enough INPs were collected during the flight to enable detection. The handling blank suspension shows a slight deviation from its water background at higher temperatures (> 253 K), but is still below the level of the UAV filter suspensions, demonstrating that the handling in the field does not significantly contribute to a contamination of INPs.

The ground filter suspensions show a spread over 7 K, showing a clear separation from the handling blank suspension as 180 well as the water background.





### 3.2 Campaign 2

For campaign 2, the setup was modified slightly (see Sect. 2.2), and in addition, the same filter was used for sampling during two consecutive flights. The flow over the filter is calculated by the mean pressures during sampling, whereas the weight is defined by the sampling time for each flight. The resulting INP concentration is shown as a function of the freezing temperature in Fig. 7

(right panel, 500,m agl) in comparison to a one-flight filter measured one day before (left panel, 400 m). By effectively doubling the flight time, more air is sampled, which in turn lowers the limit of detection, which is marked with a red horizontal line. This is especially important in clean-air environments such as the arctic (Bigg, 1996; Hatakka et al., 2003; Lohila et al., 2015; Šantl Temkiv et al., 2019). Furthermore, the two samples show two different vertical distributions for the INP concentration. While the left panel shows a very good agreement between ground and UAV filter suspension, the right panel, which was

flown one day afterwards 100 m higher, shows a difference between the two filters, especially at temperatures above 256 K. This difference highlights the importance of measuring the vertical distribution of INPs to evaluate their influence on cloud microphysics, which we demonstrate is possible with the UAV sampling system described herein.

### 3.3 Campaign 3

The modified sampler design used during campaign 3 was significantly easier to use in the field compared to the two prior

setups and also offered a higher flow due to the switch from a 25 mm diameter filter to the 47 mm diameter filter, which leads to less pressure drop across the filter. The new setup offered easier access to the filter due to it being mounted outside the fuselage. As a result of this increase in pressure downstream of the filter, the micro-diaphragm pump maintains an increased flow rate. The flight time was shorter (from about 90 minutes down to 60 minutes), due to additional weight, but this was partly compensated for by the increase in flow (from about $10.3\,\mathrm{l\,min}^{-1}$ to $11.1\,\mathrm{l\,min}^{-1}$ at 500 m asl) and the decrease in

transport losses. The INP detection limit can further be decreased by flying the same filter multiple times, therefore increasing the sampled air. By this, the onset of freezing can be observed towards higher temperatures. Even though only three flights were conducted, the setup shown in figure 1 was tested successfully with a higher flow and easier handling in the field.

### 4 Conclusions and Outlook

A lightweight and mobile unit was developed for sampling atmospheric aerosols either on a fixed-wing UAV or on the ground.

The filter-based setup was used and further improved during three field campaigns to collect INPs in low aerosol concentration environments (i.e., northern Finland, Lohila et al. (2015)) at different heights up to 1 km agl. The sampling flow was continuously measured to ensure a constant flow over the sampling period, whereas the actual flow was calculated with the average pressure during the sampling period. Ambient sensors for temperature, $T$, relative humidity, RH, and pressure, $p$, give additional information regarding current conditions and, in the case of the pressure measurement, the mass flow as well as the

height of the flight, alongside GNSS data.



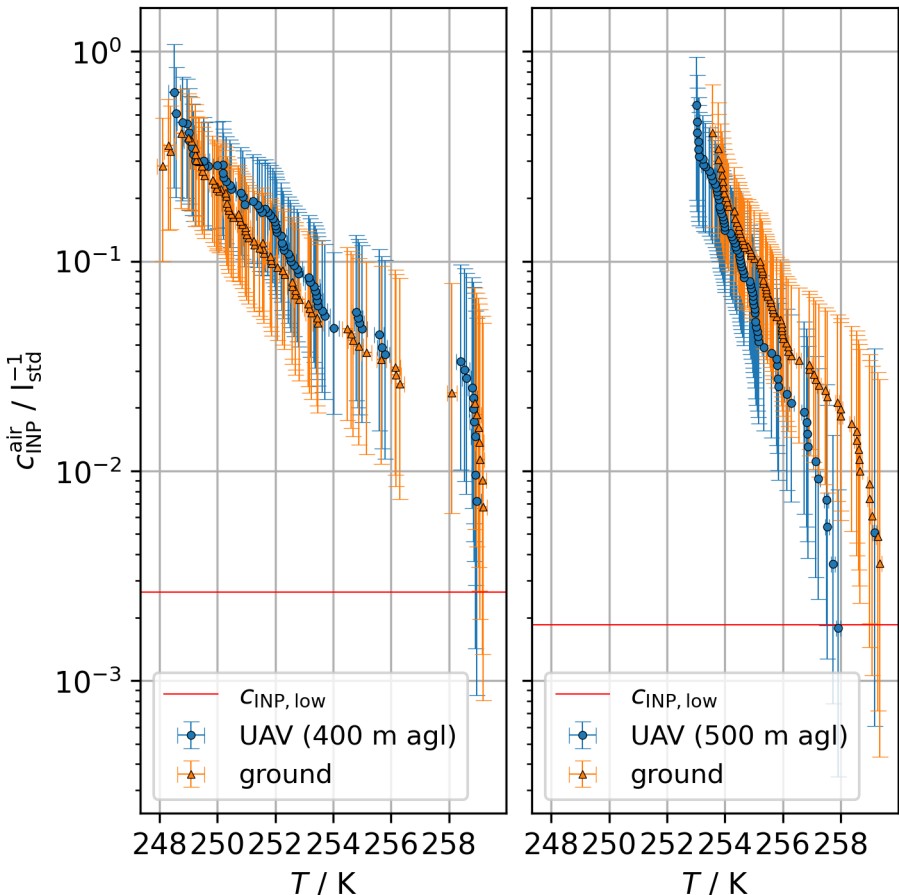

**Figure 7.** The left panel shows the INP concentration in air, $c_{\text{INP}}^{\text{air}}$, at 400 m agl as a function of the freezing temperature, $T$, for a UAV and a ground (GR) filter during campaign 2. Both filters agree very well with each other. On the right panel, the same is shown for two filters one day after at 500 m agl. This filter was flown two times, doubling its sampling time and therefore increasing the amount of air sampled (note also the decreased lower detection limit as a red horizontal line, Eq. (3)). It can be seen that lower INP concentrations can be detected, as well as a steeper freezing curve. The freezing curve does not reach the water background on the right panel. This is due to the fact that no dilution was prepared, and therefore the water background was not reached with the higher amount of INPs that can freeze a well.

The development of the setup is described in detail in this study, where the final system was optimised for field measurements in locations with low aerosol loadings. In addition, options for different operations, i.e. flying one filter multiple times to reach a lower detection limit, were tested (see Fig. 7, right panel). Typical detection limits for the setup for one flight are around $c_{\text{INP,low}}^{*} = 1 \cdot 10^{-3} \, \text{l}_{\text{std}}^{-1}$, divided by the number of flights per filter, and dependent on the flight altitude. Some flights

show a difference between the UAV filter compared to a concurrent ground-based filter, usually showing a decrease of INP concentrations at higher altitudes. This setup is able to measure the vertical distribution of INPs in a cheap and flexible way compared to aircraft or balloon-borne setups.





Future improvements will include size distribution measurements in addition to INP measurements via small, lightweight optical particle counters. The Universal Cloud and Aerosol Sounding System (UCASS) would open up this possibility (Smith et al., 2019; Girdwood et al., 2020). More measurements, for longer operation periods, in conjunction with other similar measurement platforms, i.e. ballon-borne, and at different heights, are planned and will increase knowledge about height-resolved INP concentrations inside the planetary boundary layer. These measurement periods will allow the additional use of backwards trajectories to estimate the sources of aerosols at different altitudes. A vertical distribution of INP concentrations, especially in the Arctic, could also be helpful to validate as well as complement models to connect ground- and aircraft-based measurements.

*Code and data availability.* The code for the calculation of the transport efficiency is available at https://codebase.helmholtz.cloud/alexander. boehmlaender/as_tools. The code for the creation of the plots and analysis of raw data is available from the author upon request. Data sets are available at https://radar.kit.edu/radar/en/dataset/ecljSTKjCuIoqEkr?token=sSJKlzwZKHYlpepdBzaK.

**Appendix A: Uncertainty budget of INSEKT**

The uncertainties of the preparation of the solutions as well as the washing water need to be considered during analysis. The uncertainties are calculated via propagation of uncertainty from the formula

$$c_{\text{INP,air}} = \frac{V_{\text{sol}}}{V_{\text{air}}} \frac{d}{V_{\text{well}}} \ln\left(\frac{f_{\text{l,np}}}{f_{\text{l}}}\right) . \tag{A1}$$

This results in the following formula for the variance:

$$\text{Var}(c_{\text{INP,air}}) = c_{\text{INP,air}}^2 \left[\left(\frac{\Delta V_{\text{sol}}}{V_{\text{sol}}}\right)^2 + \left(\frac{\Delta V_{\text{well}}}{V_{\text{well}}}\right)^2 + \left(\frac{\Delta V_{\text{air}}}{V_{\text{air}}}\right)^2 + \left(\frac{\Delta d_n}{d_n}\right)^2\right] . \tag{A2}$$

The uncertainty of the solution volume is dependent on its preparation and can therefore vary for different experiments. In general, it is given by the weighted sum of the pipette uncertainties

$$\frac{\Delta V_{\text{sol}}}{V_{\text{sol}}} = \frac{\sum_i w_i V_i}{V_{\text{sol}}} , \tag{A3}$$

where the weights $w_i$ are given by the fraction of pipette and solution volume. The uncertainty of the well volume is given by the electrical pipette (Eppendorf Xplorer® plus, Eppendorf, see Table A1) that is used to fill each well

$$\frac{\Delta V_{\text{well}}}{V_{\text{well}}} = \frac{6\% \cdot 50\,\mu\text{l} + 1\% \cdot 50\,\mu\text{l}}{50\,\mu\text{l}} = 7\% . \tag{A4}$$

The volume of air sampled is calculated with the measured mass flow and the duration of the sampling, therefore the corresponding uncertainty is given as

$$\frac{\Delta V_{\text{air}}}{V_{\text{air}}} = \sqrt{\left(\frac{\Delta F_{\text{std}}}{F_{\text{std}}}\right)^2 + \left(\frac{\Delta t}{t}\right)^2} , \tag{A5}$$





where the uncertainty of the time measurement is just given by the least count, i.e. half a minute. The uncertainty of the flow

can be estimated via propagation of uncertainty from the fitting function (see app. C) as

$$\Delta F_{\mathrm{std}} = \sqrt{(p\Delta m_i)^2 + (m_i\Delta p)^2 + (\Delta c_i)^2} \tag{A6}$$

with $i$ denoting the different fitting parameters and $p$ the pressure measured with the BME280 sensor.

The uncertainty of the dilution scale is given by

$$d_n = \left(\frac{V_0 + V_d}{V_d}\right)^n,$$

$$\frac{\Delta d_n}{d_n} = n\left(\frac{V_d}{V_0 + V_d}\right)\sqrt{\left(\frac{\Delta V_0}{V_0}\right)^2 + \left(\frac{V_0}{V_d}\right)^2\left(\frac{\Delta V_d}{V_d}\right)^2} \tag{A7}$$

with $n$ denoting the dilution step (i.e. 0, 1, 2, ...), $V_0$ the volume of the Nanopure water in the dilution, and $V_d$ the volume of

the washing water in the dilution. These systematic uncertainties are usually up to two orders of magnitude smaller than the

statistical uncertainties; therefore, they are omitted from the plots for simplicity.

## Appendix B: Transport efficiency

The theoretical calculations of the transport efficiency are dependent on a multitude of factors, one of them the flow regime.

The Reynolds number, $\mathrm{Re}$, is given as

$$\mathrm{Re} = \frac{v_{\mathrm{m}}d}{\nu} \approx 4100 , \tag{B1}$$

where the mean velocity, $v_{\mathrm{m}}$, is calculated with a flow of $10\,\mathrm{l\,min^{-1}}$ and a tube diameter, $d = 4 \cdot 10^{-3}\,\mathrm{m}$. The value for the

kinematic viscosity of air at $T = 273.15\,\mathrm{K}$ is calculated from the viscosity as described in Kulkarni (2011) (Eq. (2-8) and Table

2.1 therein).

The resulting transport efficiency considers diffusional losses from small particles ($d_{\mathrm{p}} \leq 20\,\mathrm{nm}$) and losses of larger particles

via sedimentation, inertial effects, and turbulent deposition (Kulkarni, 2011; Wendisch et al., 2004). The resulting efficiency as

a function of particle diameter in the respective flow regime is shown in Fig. B1. The parameters are detailed in Table B1.

**Table A1.** Uncertainties of the INSEKT pipettes (Eppendorf, 2021a, b).

| Device | Value | Systematic uncertainty | | Random uncertainty | |
|---|---|---|---|---|---|
| | | $\pm\%$ | $\pm\Delta$ | $\pm\%$ | $\pm\Delta$ |
| Eppendorf Research® plus, | 5 ml | 0.6 | 30 µl | 0.15 | 8 µl |
| violet | 2.5 ml | 1.2 | 30 µl | 0.25 | 6 µl |
| Eppendorf Research® plus, blue | 100 µl | 3 | 3 µl | 0.6 | 0.6 µl |
| Eppendorf Xplorer® plus, blue | 50 µl | 6 | 3 µl | 1 | 0.5 µl |





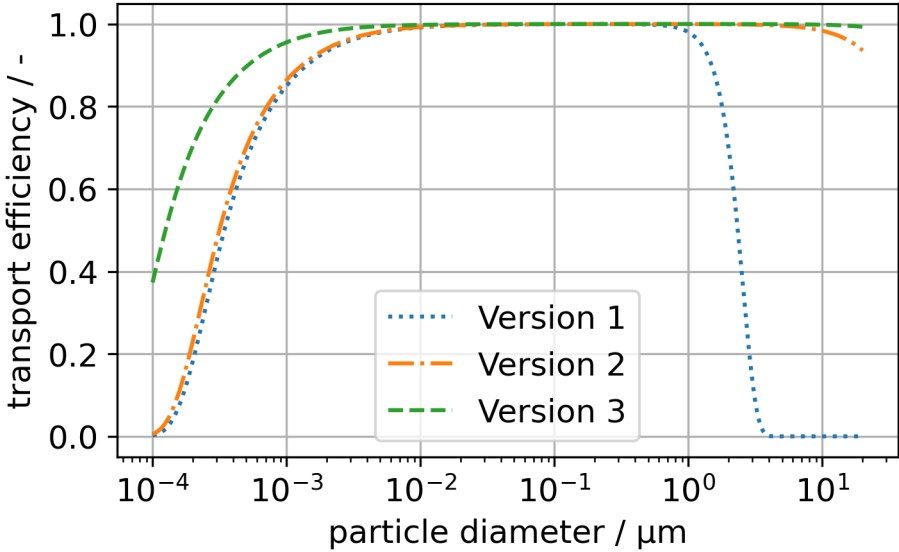

**Figure B1.** The total transport efficiency is calculated for all three setups used in the three campaigns. Especially larger particles have a larger transport efficiency due to the removal of bends for the second setup and the increase in flow for the third setup. The decrease in the flow path length benefits the collection of small particles due to the decreased diffusional losses.

**Appendix C: Flow calculation**

The micro-diaphragm pump consists of two diaphragms that are shifted by about 180° and therefore produce a flow that follows an absolute sinus curve. The maximal values of this sinus are higher than the rated full scale of the flow meter. Therefore, the flow meter used in the setup will always underestimate the flow. The flow meter is still useful since one can determine if there was any flow over the filter or if there was any change during filter sampling, which can occur on a UAV due to vibrations, for

**Table B1.** Parameters for the calculation of the transport efficiency in all three setups.

| Parameter | 1st setup | 2nd setup | 3rd setup |
|---|---|---|---|
| Tube length / m | $172 \cdot 10^{-3}$ | $145 \cdot 10^{-3}$ | $20 \cdot 10^{-3}$ |
| Tube diameter / m | | $6 \cdot 10^{-3}$ | $4 \cdot 10^{-3}$ |
| Flow / $l\,min^{-1}$ | | 10 | 11 |
| Ambient pressure / Pa | | 95000 | |
| Ambient temperature / K | | 273.15 | |
| Inclination angle / ° | | 0 | |
| Bend angle / ° | 90 | | — |
| Number of bends / — | 2 | | 0 |



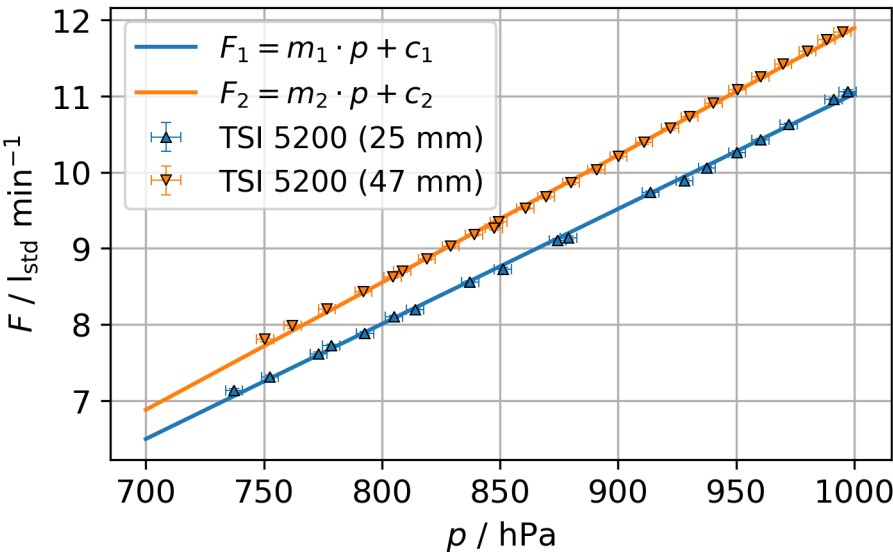

**Figure C1.** The flow was measured with a flow meter (TSI 5200, TSI) and created by the micro-diaphragm pump at different pressures. The pressure was measured by a pressure sensor (VD85, Thyracont) in line with the flow upstream of the filter. The uncertainty of the flow is obtained from the standard deviation of the mean during measurement. The uncertainty on the pressure is taken from the data sheet of the respective sensors (VD85, 0.3 % of full scale, Thy (2021)).

instance. The pump was tested in the lab in combination with a needle valve, a pressure sensor (VD85, Thyracont), and a flow meter (TSI 5200, TSI) to measure the mass flow at different pressures. Figure C1 shows the linear nature of the dependence
of the flow on the pressure in the observed pressure range. Using the pressure from the VD85 sensor, orthogonal distance regression (ODR, Boggs and Rogers (1990)) was used to estimate the flow during a flight in dependence of the pressure.

*Author contributions.* AB wrote this manuscript and performed the data and filter analysis. LL and OM provided scientific discussion as well as proofreading of the whole manuscript. DB and KD contributed to the manuscript and were responsible for flying the fixed-wing during all campaigns. ZB and MB took care of flying and filter changing during the spring 2021 campaign. JK was responsible for the design and
275 printing of the 3D pieces. TL has written the LabVIEW code for INSEKT and assisted during data extraction.

*Competing interests.* The authors declare that they have no conflict of interest.

*Acknowledgements.* Map data copyrighted OpenStreetMap contributors and available from https://www.openstreetmap.org. We gratefully acknowledge support from the technical and engineering team members at IMK-AAF, in particular Jens Nadolny. This work received financial



support through the Helmholtz program "The Atmosphere in Global Change", and was part of a transnational access project that was supported by the European Commission under the Horizon 2020 – Research and Innovation Framework Programme, H2020-INFRAIA-2020-1, ATMO-ACCESS Grant Agreement number: 101008004.





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
