# Peer review of "A novel aerosol filter sampler for measuring the vertical distribution of ice-nucleating particles via fixed-wing uncrewed aerial vehicles"

_Atmospheric Measurement Techniques, 2024_

## Author Comment (AC1)

**Author response to RC1**

December 22, 2024

We are grateful for the helpful comments and suggestions from the reviewer. Below the reviewers comments are in blue with our responses in black directly below.

The manuscript describes the application of a newly deployed mobile filter sampling system, operating on an uncrewed aerial vehicle (UAV). The development and improvement of the sampling technique over three subsequent measurement campaigns is well documented. Also, the text is well written, everything is well described.

However, occasionally statements need correction, and in some parts additional clarifications are needed, as specified in the general comments. Also, there is one mayor criticism I have that may well go beyond this study: For testing new equipment, test flights were done in northern Finland. It is not clear why they were not done at the home institute, and instead a lot of travel and transport of both people and equipment was done to get these results. This does not diminish the quality of this work, but should at least be considered in future developmental work.

After below comments will have been addressed, I can support publication in AMT.

The authors agree that the reduction in travel and transport of both people and equipment should be a common goal for developmental work, especially in regard to climate change. We do see this as a valid criticism and want to offer some answers on why we choose the area in northern Finland.

We have a long standing collaboration with the Finnish Institute of Meteorology (FMI), having participated in previous Pallas Cloud Experiment (PaCE) campaigns. Their experience in operating UAVs was highly beneficial for the success of the development of our filter-based INP sampling setup.

Regulations for the use of UAVs are very restrictive in Germany and in general in Europe. This is one major hurdle for the operation of UAVs. The existence of a temporary danger zone around the Sammaltunturi station in Finland allows a lot of flexibility. The collaboration with the FMI and their offer to use this temporary danger zone for our testing and validation experiments was highly beneficial and added to the overall success of this project.
* * *
**1 General comments**

Line 69: "regions with low aerosol concentrations, such as northern Finland" sounds wrong. An annual cycle of the Arctic aerosol has been understood for some time now. Concerning size distributions or number concentrations, you can check out e.g. Tunved et al. (2013) or Freud et al. (2017). Arctic haze contributes particles in winter time, new particle formation causes elevated particle concentrations in summer time. Concerning INP concentrations, it has also been understood that there is an annual cycle with high concentrations possible already in spring and advancing well into autumn (see e.g., Creamean et al., 2018, Wex et al., 2019; Porter et al., 2023; Schneider et al., 2021, with the latter one from your institute). Therefore, the assumption of low INP concentrations in norther Finland is somewhat amazing and wrong. Please correct. (BTW: Check your literature list. For Schneider et al. (2021), I saw that the doi refers to the discussion paper. – I did not check the rest.)

We thank the reviewer for this comment. We agree that our statement is too strong and not precise enough. Our campaigns were either conducted during spring or autumn near the Sammaltunturi Global Atmosphere Watch (GAW) site (67°58'N 24°7'E, 565 m above sea level (asl), northern Finland, Lohila et al. 2015). Sammaltunturi is located in a subarctic environment above the tree line around 180 km north the Arctic circle. While the total number concentration at Sammaltunturi can reach higher concentrations ($< 10\,000\,\text{cm}^{-3}$), this is as the reviewer states only the case for summertime. Spring and autumn concentrations are typically one order of magnitude less, see for example Lohila et al. 2015.

[Figure]

Figure 1: Figure 13 from Asmi et al. 2011.

We also want to add that while Arctic haze does contribute during winter and spring time in the Arctic, the area around Sammaltunturi is typically considered subarctic and to the authors knowledge is generally not representative for Arctic haze compared to stations like Zeppelin that are typically inside the polar dome during winter and spring (e.g., Asmi et al. 2011, see fig. 13 therein and fig. 1). Tunved et al. 2013 describe the aerosol measurements from the Zeppelin station, which is very much representative for an Arctic station that is affected by Arctic haze. Freud et al. 2017 reports measurements "from five sites around the Arctic Ocean (Alert, Villum Research Station – Station Nord, Zeppelin, Tiksi and Barrow)" but not from Sammaltunturi itself. Schmale et al. 2022 analyzed aerosol measurements of 10 Arctic observatories, one of them being Sammaltunturi (denoted as Pallas). Sammaltunturi is much more influenced by anthropogenic emissions from Europe and Asia and by biogenic emissions from the boreal forest compared to other Arctic stations, such a Zeppelin (Schmale et al. 2022).

Asmi et al. 2011 compares different stations in mainland Europe and subarctic and Arctic locations on aerosol concentration for a two-year-period. Table 4 of Asmi et al. 2011 shows the mean number concentration of Sammaltunturi (denoted as Pallas) as $198\,\mathrm{cm}^{-3}$, $387\,\mathrm{cm}^{-3}$ and $211\,\mathrm{cm}^{-3}$ for the size ranges $30\,\mathrm{nm}$ to $50\,\mathrm{nm}$, $>50\,\mathrm{nm}$ and $>100\,\mathrm{nm}$, respectively. Only two stations report lower concentrations, them being a high-altitude station (Jungfraujoch in the Swiss Alps) and a high Arctic station (Zeppelin on Svalbard). This is even more pronounced when looking at the lower quantiles. We therefore think our original statement is not wrong. We agree that we need to be more careful with this statement and have therefore added the following to the mentioned sentence:

old  This setup proved the technical feasibility of measuring INPs with sample times between 45 and 90 minutes in regions with low aerosol concentrations, such as northern Finland.

new  This setup proved the technical feasibility of measuring INPs with sample times between 45 and 90 minutes in regions with low aerosol concentrations, such as the area around Sammaltunturi in northern Finland during the measurement periods in spring and autumn.

In reference to INP measurements conducted by our institute in southern Finland (Hyytiälä, Smear II, Schneider et al. 2021), we would like to add that the measurement location is quite different to Sammaltunturi and that the mean number concentration in Hyytiälä (Smear II, southern Finland) is higher compared to Sammaltunturi: $345\,\mathrm{cm}^{-3}$, $1053\,\mathrm{cm}^{-3}$ and $450\,\mathrm{cm}^{-3}$ for the same size ranges (Asmi et al. 2011). This results in factors of 1.74, 2.72 and 2.13, respectively. If we consider a similar ice-active fraction of the aerosols, this would equal that INP concentrations measured at Sammaltunturi are lower by a similar factor.

To conclude: we do not make the statement that Sammaltunturi generally features low INP concentrations, but we understand that it might be implied from our wording. We have therefore added an additional sentence:

new  While low aerosol concentrations might also lead to lower INP concentrations, this is not always true, since some aerosols might be much more ice active than others and therefore contribute disproportionally to the INP concentration (e.g., Hoose and Möhler 2012).

We are grateful to the reviewer for reviewing the literature and have changed the reference of Schneider et al. 2021 to accurately refer to the finalized publication. We have also checked the rest of the literature to make sure there are no other mistakes to the best of our knowledge.
* * *
Line 134-135: Why do you subtract the water background, but not, instead, the background from the field handling blank? By this, you create some in-between state with the background partially, but not fully, subtracted.

To the authors' knowledge no consensus has been reached in the INP community on how to handle handling blank filters. We have discussed if we would subtract the handling blank from the filter, but have ultimately decided not to, because of the following reasons: the water is subtracted for the frozen fraction, since one is unable to associate it with any amount of air volume sampled. The handling blank is analysed in the same way as the actual filter samples and the handling blank is typically very close to its water background. This introduces issues since the water background is slightly different for each experiment. Therefore, it is also slightly different for the handling blank. For these reasons, we use the handling blank to assess our handling procedure, but do not subtract its frozen fraction from the original samples. This way we can flag filters as invalid, if they were taken after a handling blank filter whose aerosol suspension shows a contamination.
* * *
Line 169: Why are you talking about campaign 2 here at the beginning of the description of campaign 1? Did you not take field handling blanks during campaign 1? Then better mention that explicitly.

We do not report any results from campaign 1. Campaign 1 was a proof-of-concept study and meant to test the technical feasibility of the general setup. One major objective of campaign 1 was the investigation of the INP concentration obtained from such a filter-based UAV setup with relatively short sampling times with a focus on area with typically and relatively low aerosol concentrations. For campaign 2 we report more substantial and significant results, where we also focused on taking handling blanks.

We would like to show some results from campaign 1 to the reviewer (see fig. 2), showcasing that we did detect INPs with the first setup, but we would like to focus our results on the second setup:
* * *
Line 174 and Fig. 6: Do you have any idea why the handling blank on the UAV is higher than that on ground? And is it right that there was only one handling blank for UAV and one for the ground? Then please mention this explicitly.

There is no identified reason why the handling blank on the UAV is higher than on the ground. The frozen fraction of the handling blank for the UAV is still below the frozen fraction of the UAV filters. Unfortunately we only took one handling blank filter for this setup. We have changed the following sentence to mention is explicitly:

[Figure]

Figure 2: Frozen fraction of an aerosol sample collected during campaign 1 on the ground (panel (a)) and on-board of a UAV (panel (b)). Both suspensions show a clear separation from the water background (denoted as Nanopure).

old To demonstrate the scientific feasibility, *the inverse of the liquid fraction,* the frozen fraction, of the UAV and ground filters are compared to their respective handling blank filters taken during campaign 2.

new To demonstrate the scientific feasibility, the frozen fraction of the UAV and ground filters are compared to their respective handling blank filters taken during campaign 2. Only one handling blank filter was taken for each setup.

Note that the part in *italic* was changed due to comments from reviewer 2.
* * *
Line 146 ff: About the detection limit: The argument in this paragraph is too complicated and can be simplified a lot. The complication is by first assuming that in the whole batch there should be one INP, then getting a value for that, but then saying (correctly), that it is not the whole batch that is examined, but only one droplet. The number given in the text for the whole batch is useless, and no number (or range) is given here for the results of equation (3), which really is the lower detection limit. (I only later on found one value in line 214.) Instead, it could just be said that the detection limit comes from equation (2) with the assumption of one frozen droplet (i.e., one of the 64 droplets frozen, or f_l = 63/64 = 0.984). That will be the minimum value you get as a concentration, and is in the same order of magnitude than the value you give in line 214. Modify / simplify this paragraph!

We are thankful for the careful reading of this section from the reviewer. The reviewer is discussing the resolution of the instrument, which is of course correct in the sense that our minimal resolution is directly tied to the amount of simulated droplet. Our assumption is related to the total volume of the water used to wash the aerosol of the filter. There are 64 wells filled with $50\,\mu\mathrm{L}$ each, which equals a total of $3200\,\mu\mathrm{L} = 3.2 \times 10^{-3}\,\mathrm{L}$. Our assumption is now that the experiment does detect an INP if there is one INP in any of those wells. This signal would be different to the pure water signal. Combining this with a volume of the washing water of $5 \times 10^{-3}\,\mathrm{L}$, results in a factor of roughly 1.56. The actual detection limit is therefore by this factor higher than the simple assumption of the inverse of the sampled air volume. We update this paragraph to make our intention more clear:

old The lower detection limit can be estimated by the condition that a single INP has to exist to initiate freezing, i.e. for a sampled volume of $500\,\mathrm{L}_{\mathrm{std}}^{-1}$, the rough estimate for the lower detection limit for the INP concentration is $c_{\mathrm{INP,low}}^* = 2 \times 10^{-3}\,\mathrm{L}_{\mathrm{std}}^{-1}$. This detection limit also depends on the analysis, since the whole suspension is not used for one analysis.

new The lower detection limit can be estimated by the condition that a single INP has to exist to initiate freezing, i.e. for a sampled volume of $V = 500\,\mathrm{L_{std}^{-1}}$, the rough estimate for the lower detection limit for the INP concentration is $c_{\mathrm{INP,low}}^{*} = V^{-1} = 2 \times 10^{-3}\,\mathrm{L_{std}^{-1}}$. This detection limit also depends on the analysis, since only a fraction of the whole suspension is used for one analysis.
* * *
Figure 7: In the caption, you refer to the steepness of the sampling curve such that the impression arises that the steepness of the curve depends on the sampling time. But such a dependency does not exist! The steepness of the curve reflects which INPs are there in which concentrations. Each INP has its characteristic freezing (or ice-activation) temperature (in the non-stochastic approach you are using here). And the curve shows how many INPs are there that are ice-active at the different temperature. So the shape of the curve has nothing to do with the sampling time. You can only argue, as you do, that by increasing the volume of sampled air you also increase the maximum temperature at which you get data (by decreasing the minimum detection limit), and that you may see different parts of the curve. But the curve at a certain temperature range will be the same, independent how long you sample. Otherwise the whole measurement approach would not make sense. And that increase in maximum temperature is not much, as doubling the sampling time only halves concentration, which is not much gain in terms of temperature at which data is available.

Bottom line: Modify the caption of Fig. 7 (and elsewhere in the text if needed) such that this wrong impression of a connection between sampling time and steepness of the curve does not appear any more.

We thank the reviewer for their careful reading. We did not want to raise the impression that the steepness of the curve is associated with the sampling time. We removed the discussion of the steeper curve and modified the original sentence to remove any ambiguity:

old It can be seen that lower INP concentrations can be detected, as well as a steeper freezing curve.

new It can be seen that lower INP concentrations can be detected due to the increased sampling time.
* * *
Line 187: Again, as above, the Arctic is not on its own a region with clean air (see e.g., Arctic haze) or low INP concentrations (see my remark for line 69). Please reword this sentence.

See our answer to your previous comment. We have moved this sentence into the Conclusion and Outlook section and added an additional part to explain why the Arctic is especially interesting for measuring the vertical distribution of INPs:

new This is the case especially in the Arctic due to its characteristically stratified atmosphere (Graversen et al. 2008).
* * *
Line 191: "This difference highlights the importance of measuring the vertical distribution." While I agree that differences in INP concentrations are to be expected with altitude, it becomes less clear in the presented data than what I would have expected. It would be more correct to state that while you do not present data showing big differences between ground and UVA, stronger differences can be expected and measuring the vertical distribution is important.

We want to highlight that this setup is able to measure INP concentrations at different altitudes and is therefore able to detect differences, but based on the data presented in this paper no conclusions about potential differences can be drawn. We have adjusted the following sentence:

old This difference highlights the importance of measuring the vertical distribution of INPs to evaluate their influence on cloud microphysics, which we demonstrate is possible with the UAV sampling system described herein.

new This difference, albeit small, highlights the importance of measuring the vertical distribution of INPs to evaluate their influence on cloud microphysics. The data presented in this study demonstrate that such investigation is possible with the UAV sampling system described herein.

We also would like to add that we did additional and more comprehensive measurements during a Pallas Cloud Experiment (PaCE) campaign in autumn 2022, which will be published in the future.
* * *
Given that we are living in times of climate change and should rather watch our resources, a question arises, overall: Why were these tests not done at home, but instead, for a few research fights, a lot of travel and transport was done?

We agree with the reviewer and would like to refer to our answer at the beginning of this response. Each and every travel should be carefully evaluated and a decision should be made with the consequences and the benefits evaluated against each other.
* * *
Also: How did you realize that the different inlets performed differently? Only by modeling? Please add this information to the text.

We have only used well-established theoretical equations for the estimation of the transport efficiency, which can be found in standard text books (e.g. Kulkarni 2011). To reflect this better in the text, we changed the following sentence:

old This change enhanced the sampling efficiency of larger aerosol particles due to fewer bends...

new This change enhanced the theoretical sampling efficiency of larger aerosol particles due to fewer bends...
* * *
 You claim that INP concentrations are "usually showing a decrease of INP concentrations at higher altitudes". This cannot be seen from the data in Fig. 7, given the agreement within measurement uncertainty in both panels and the fact that data obtained from the UAV is even higher than data collected at ground in the left panel. Either show different data in Fig. 7 if you have them, or correct the text (here and throughout the manuscript) accordingly.

We have adjusted this sentence from the manuscript. For these first results, we do see a general trend, but we agree that we need a larger amount of data to create a reliable statistic and give concrete conclusions. We changed the earlier part:

old Some flights show a difference between the UAV filter compared to a concurrent ground-based filter, usually showing a decrease of INP concentrations at higher altitudes.

new Some flights show a difference between the UAV filter compared to a concurrent ground-based filter, but based on the current dataset it is not possible to draw a statistically relevant conclusion.
* * *
 In this part, you give future plans instead of highlighting why your approach was a useful addition. The impression arises that you found that your new approach is not very good. You can leave that as it is, but maybe it diminishes the value of your work too much.

We have adjusted this sentence:

old Future improvements will include size distribution measurements in addition to INP measurements via small, lightweight optical particle counters.

new Future experiments will include size distribution measurements in addition to INP measurements via small, lightweight optical particle counters.

And also added two new sentence to the previous paragraph to better highlight the usefulness of our approach:

old

new The current work provides a solid foundation for understanding INP concentrations in varied atmospheric conditions. Additional measurements will further enhance the statistical robustness and reliability of our findings.

Line 223: You can already use backwards trajectories for your own UAV measurements, so the question arises why you did not do that if you deem them so valuable, here.

We hope to combine different approaches in the future for identifying air mass and aerosol origin. The focus of this study is on the technical aspect of the measureeent setup, and gives an overview of the improvements made to refine our method. We aim at including more information about aerosol characteristics and origin, e.g., by using FLEXPART (Flexible Particle Dispersion Model, Pisso et al. 2019) in future publications. This is one major aspect of a field campaign that we participated in during autumn 2022 (Pallas Cloud Experiment 2022) which will be published soon.

Line 224: Why especially in the Arctic? Please justify (half a sentence can be enough) if you want to make this claim.

We removed this part of the sentence and added it as a new sentence with a justification:

old A vertical distribution of INP concentrations, especially in the Arctic, could also be helpful to validate as well as complement models to connect ground- and aircraft-based measurements.

new A vertical distribution of INP concentrations could also be helpful to validate as well as complement models to connect ground- and aircraft-based measurements. This is the case especially in the Arctic due to its characteristically stratified atmosphere (Graversen et al. 2008).

**2 Specific and editorial comments**

Line 3: Add "in Finland" to the site description.

We added this part.

old A mobile sampler for collecting aerosol particles on an uncrewed aerial vehicle (UAV) was developed and tested during three consecutive Pallas Cloud Experiment campaigns in the vicinity of the Sammaltunturi Global Atmosphere Watch site (67°58' N, 24°7' E, 565 m above sea level).

new A mobile sampler for collecting aerosol particles on an uncrewed aerial vehicle (UAV) was developed and tested during three consecutive Pallas Cloud Experiment campaigns in the vicinity of the Sammaltunturi Global Atmosphere Watch site (67°58' N, 24°7' E, 565 m above sea level) in Finland.

Line 26-28: This sentence is totally correct, but not needed in the context of this work. Consider deleting it, although I will not force you to do it.

We would like to keep this part in as an additional motivation of our research.

Line 111: wrong format for citing Schneider et al. (2020). Try using citep[text preceeding citation][text following citation]label. (If this won't work for AMT, there will be a similar way, maybe with <...> instead of [...] ).

We are assuming the reviewer means that we should remove the brackets around the year of the citation. We changed the original command from
`INSEKT, see e.g.~\citet{Schneider2020}`
to
`\citep[INSEKT, see e.g.,][]{Schneider2021}`, which results in:

old Schneider et al. (2020)

new Schneider et al. 2021

We have also adjusted this for the following citations to be consistent:

old The newly developed UAV-based aerosol sampler was used and further developed during three Pallas Cloud Experiment campaigns, close to the Sammaltunturi Global Atmosphere Watch (GAW) site (67°58'N 24°7'E, 565 m above sea level (asl), northern Finland, Lohila et al. (2015)), which took place during autumn 2020, spring 2021, and autumn 2021.

new The newly developed UAV-based aerosol sampler was used and further developed during three Pallas Cloud Experiment campaigns, close to the Sammaltunturi Global Atmosphere Watch (GAW) site (67°58'N 24°7'E, 565 m above sea level (asl), northern Finland, Lohila et al. 2015), which took place during autumn 2020, spring 2021, and autumn 2021.

old The filter-based setup was used and further improved during three field campaigns to collect INPs in low aerosol concentration environments (i.e., northern Finland, Lohila et al. (2015)) at different heights up to 1 km agl.

new The filter-based setup was used and further improved during three field campaigns to collect INPs in low aerosol concentration environments (i.e., northern Finland, Lohila et al. 2015) at different heights up to 1 km agl.
* * *
Line 183-184: Which weight are you talking about, here? And is this important?

We are calculating the flow with a weighted mean of the two flights. The weight here is the sampling time and not associated with any physical weight. We have added the following statement to make this clearer:

old The flow over the filter is calculated by the mean pressures during sampling, whereas the weight is defined by the sampling time for each flight.

new The flow over the filter is calculated by the mean pressures during sampling. The actual flow is the weighted arithmetic mean, where the weight is defined by the sampling time for each flight.
* * *
Line 187: Names are always capitalized -> "Arctic".

We have changed this.

old This is especially important in clean-air environments such as the arctic (Bigg 1996; Hatakka et al. 2003; Lohila et al. 2015; Šantl-Temkiv et al. 2019).

new This is especially important in clean-air environments such as the Arctic (Bigg 1996; Hatakka et al. 2003; Lohila et al. 2015; Šantl-Temkiv et al. 2019).

Due to a comment from the second reviewer, we have also moved this sentence to the Conclusion and Outlook section.
* * *
Equation A2: Explicitly state also here, that Ddn is a specific dilution.

We have added this statement and made a reference to equation A7.

old This results in the following formula for the variance:

$$\mathrm{Var}(c_{\mathrm{INP,air}}) = c_{\mathrm{INP,air}}^2 \left[ \left( \frac{\Delta V_{\mathrm{sol}}}{V_{\mathrm{sol}}} \right)^2 + \left( \frac{\Delta V_{\mathrm{well}}}{V_{\mathrm{well}}} \right)^2 + \left( \frac{\Delta V_{\mathrm{air}}}{V_{\mathrm{air}}} \right)^2 + \left( \frac{\Delta d_n}{d_n} \right)^2 \right]. \tag{A2}$$

new This results in the following formula for the variance:

$$\mathrm{Var}(c_{\mathrm{INP,air}}) = c_{\mathrm{INP,air}}^2 \left[ \left( \frac{\Delta V_{\mathrm{sol}}}{V_{\mathrm{sol}}} \right)^2 + \left( \frac{\Delta V_{\mathrm{well}}}{V_{\mathrm{well}}} \right)^2 + \left( \frac{\Delta V_{\mathrm{air}}}{V_{\mathrm{air}}} \right)^2 + \left( \frac{\Delta d_n}{d_n} \right)^2 \right], \tag{A2}$$

where $d_n$ denotes a specific dilution (see Eq. (A7)).

Line 265: As this part is quite separate from the main text, specify here which flow meter you are referring to here, with "the flow meter".

We have added this information in the Appendix to make it clear which flow meter we are referring to:

old The maximal values of this sinus are higher than the rated full scale of the flow meter.

new The maximal values of this sinus are higher than the rated full scale of the flow meter (SFM4100).

**References**

Asmi, A. et al. (2011). "Number size distributions and seasonality of submicron particles in Europe 2008–2009". In: *Atmos. Chem. Phys.* 11.11, pp. 5505–5538. DOI: 10.5194/acp-11-5505-2011.

Bigg, E. K. (1996). "Ice forming nuclei in the high Arctic". In: *Tellus Ser. B* 48.2, pp. 223–233. DOI: 10.3402/tellusb.v48i2.15888.

Freud, E. et al. (2017). "Pan-Arctic aerosol number size distributions: seasonality and transport patterns". In: *Atmospheric Chemistry and Physics* 17.13, pp. 8101–8128. DOI: 10.5194/acp-17-8101-2017.

Graversen, R. G., T. Mauritsen, M. Tjernström, E. Källén, and G. Svensson (2008). "Vertical structure of recent Arctic warming". en. In: *Nature* 451.7174, pp. 53–56. DOI: 10.1038/nature06502.

Hatakka, J. et al. (2003). "Overview of the atmospheric research activities and results at Pallas GAW station". In: *Boreal Environment Research* 8.

Hoose, C. and O. Möhler (2012). "Heterogeneous ice nucleation on atmospheric aerosols: a review of results from laboratory experiments". In: *Atmos. Chem. Phys.* 12.20, pp. 9817–9854. DOI: 10.5194/acp-12-9817-2012.

Kulkarni, P. (2011). *Aerosol measurement : principles, techniques, and applications.* Ed. by P. Kulkarni, P. A. Baron, and K. Willeke. Hoboken, N.J: Wiley. ISBN: 9780470387412.

Lohila, A. et al. (2015). "Preface to the special issue on integrated research of atmosphere, ecosystems and environment at Pallas". In: *Boreal Environment Research.* Vol. 20. 4, pp. 431–454.

Pisso, I. et al. (2019). "The Lagrangian particle dispersion model FLEXPART version 10.4". In: *Geoscientific Model Development* 12.12, pp. 4955–4997. ISSN: 1991-9603. DOI: 10.5194/gmd-12-4955-2019.

Šantl-Temkiv, T. et al. (2019). "Biogenic Sources of Ice Nucleating Particles at the High Arctic Site Villum Research Station". In: *Environmental Science & Technology* 53.18, pp. 10580–10590. DOI: 10.1021/acs.est.9b00991.

Schmale, J. et al. (2022). "Pan-Arctic seasonal cycles and long-term trends of aerosol properties from 10 observatories". In: *Atmospheric Chemistry and Physics* 22.5, pp. 3067–3096. ISSN: 1680-7324. DOI: 10.5194/acp-22-3067-2022.

Schneider, J. et al. (2021). "The seasonal cycle of ice-nucleating particles linked to the abundance of biogenic aerosol in boreal forests". In: *Atmospheric Chemistry and Physics* 21.5, pp. 3899–3918. ISSN: 1680-7324. DOI: 10.5194/acp-21-3899-2021.

Tunved, P., J. Ström, and R. Krejci (2013). "Arctic aerosol life cycle: linking aerosol size distributions observed between 2000 and 2010 with air mass transport and precipitation at Zeppelin station, Ny-Alesund, Svalbard". In: *Atmos. Chem. Phys.* 13.7, pp. 3643–3660. DOI: 10.5194/acp-13-3643-2013.

---

## Author Comment (AC2)

**Author response to RC2**

December 22, 2024

We are grateful for the helpful comments and suggestions from the reviewer. Below the reviewers comments are in blue with our responses in black directly below.

The authors report the development and application of an aerosol filter sampler that fits onboard fixed-wing UAVs. This technique paper will be worth publishing if it includes the discussion & implication of ground INP vs. UAV-collected INP results in Sect. 3. This reviewer wishes to see what the authors learned and what readers should be aware of in Sect. 3. Also, any INP data from Campaigns 1 and 3? If there are any, it would be worth reporting the change/improvement in INP results from the 1st setup to the 3rd one.

Unfortunately, we are unable to directly report an improvement in the quality of the INP conentration measurements from the setup improvements. This is - in part - because we do not have an INP concentration reference instrument, that we could compare our data to. Neither did we have the opportunity to measure the aerosol size distribution for the different setups directly, but had to on well-established theoretical estimations of the collection efficiency as presented in Appendix B. This paper focuses on the scientific feasibility of the described setup. The authors would hesitate to draw any conclusion from the data obtained during campaign 1 to 3, as not enough experiments were conducted. In addition, we want to add that an intensive field campaign (Pallas Cloud Experiment 2022) was conducted during autumn 2022 at the same location, featuring INP reference instrumentation on the nearby Sammaltunturi station for a longer time period of one month. Data from this campaign will be published in a special issue from Earth System Science Data.
* * *
L1: for the collection of –> for collecting. L2: deployed –> tested

We have corrected this part.

old  A mobile sampler for the collection of aerosol particles on an uncrewed aerial vehicle (UAV) was developed and deployed during three consecutive Pallas Cloud Experiment campaigns in the vicinity of the Sammaltunturi Global Atmosphere Watch site (67°58' N, 24°7' E, 565 m above sea level).

new  A mobile sampler for collecting aerosol particles on an uncrewed aerial vehicle (UAV) was developed and tested during three consecutive Pallas Cloud Experiment campaigns in the vicinity of the Sammaltunturi Global Atmosphere Watch site (67°58' N, 24°7' E, 565 m above sea level) *in Finland*.

Note that the part in *italic* was changed due to comments from reviewer 1.
* * *
L3-5: The sentence is running long. Maybe "The sampler is composed of Nuclepore … analyzed for INP abundance offline as a function of temperature using a cold stage assay." Abstract does not need to provide the full name of a developed technique, such as INSEKT. It can be mentioned in the main manuscript.

Changed the sentence to:

old  The sampler is designed to collect aerosol particles onto Nuclepore filters, which are subsequently analysed for the temperature-dependent number concentration of ice-nucleating particles of the sampled aerosol with the Ice Nucleation Spectrometer of the Karlsruhe Institute of Technology (INSEKT).

new The sampler is designed to collect aerosol particles onto Nuclepore filters, which are subsequently analysed for the temperature-dependent number concentration of ice-nucleating particles of the sampled aerosol using a freezing assay.

We would like to use freezing assay instead of cold stage assay to separate INSEKT from a cold stage.
* * *
L5-6: This setup is… This sentence is not adding much value – sounds like a sales brochure statement. This review suggests the authors exclude this sentence.

We have removed this sentence.

old This setup is an easy and flexible way to connect INP concentration measurements with cloud microphysics.

new
* * *
L7-8: 1 hour to what upper limit time? More than 1.5 hours means? Please define it in L7-8 in a simple manner.

The upper limit for the sampling time is defined by the flight time at the designated altitude of the UAV. This flight time depends on the ambient conditions and also on the designated height. We performed flights that lasted more than 100 minutes, but these were ideal conditions and a more typical upper limit is 90 minutes. We have changed our statement accordingly:

old The total flight time ranges from 1 hour to more than 1.5 hours, depending on environmental conditions.

new The total flight time ranges from 60 min to around 100 min, depending on environmental conditions.
* * *
L9-10: The authors complement a shorter flight with a relatively large air sampling flow rate, which is good. But how did the author make sure the sampling flow is laminar? Any simulation done?

The flow regime was estimated by calculating the Reynolds number taking into account a flow of $10 \, \mathrm{l\,min^{-1}}$ and a tube diameter of $4 \times 10^{-3} \, \mathrm{m}$, resulting in a Reynolds number of roughly 4100, resulting in a turbulent regime. The transport efficiency was calculated with this in consideration as well. No simulation were done apart from the ones described in the paper and in the Appendix. The following is the relevant section of the Appendix:

The theoretical calculations of the transport efficiency are dependent on a multitude of factors, one of them the flow regime. The Reynolds number, Re, is given as

$$\mathrm{Re} = \frac{v_\mathrm{m} d}{\nu} \approx 4100 \ , \tag{B1}$$

where the mean velocity, $v_\mathrm{m}$, is calculated with a flow of $10 \, \mathrm{l\,min^{-1}}$ and a tube diameter, $d = 4 \times 10^{-3} \, \mathrm{m}$.
* * *
L35: … have become one of the focuses for airborne aerosol measurements …

We want to highlight here that UAV are also of focus for ambient measurements, for example measuring similar parameters as on radiosondes. We changed the sentence accordingly:

old Recently, uncrewed aerial vehicles (UAVs) have become a focus for atmospheric measurements (e.g., Bärfuss et al. 2018; Lampert et al. 2020; Marinou et al. 2019; Villa et al. 2016; Yu et al. 2017).

new Recently, uncrewed aerial vehicles (UAVs) have become one of the focuses for atmospheric measurements (e.g., Bärfuss et al. 2018; Lampert et al. 2020; Marinou et al. 2019; Villa et al. 2016; Yu et al. 2017).
* * *
L74-75: Right now this sentence sounds like a KNF pump was only employed for your final (3rd) setup in fall 2021. Is that the case? Then, how were aerosol particles collected on a filter without a pump in the 1st and 2nd setup?

The same pump was used for all three setups. To reflect this in the manuscript, we have changed the sentence:

old In our final setup, the payload contains a micro-diaphragm pump (NMP850.1.2KPDC-B HP, KNF), which provides a flow of $15\,\mathrm{l\,min^{-1}}$ at standard conditions (KNF).

new In each setup, the payload contains a micro-diaphragm pump (NMP850.1.2KPDC-B HP, KNF), which provides a flow of $15\,\mathrm{l\,min^{-1}}$ at standard conditions (KNF).
* * *
We agree and have changed this figure to an actual picture of the mounted filter holder with the individual components.

[Figure]

Figure 1: 3D printed parts for the filter holder mounting to the plane. From left to right: filter holder mount, mount base, mount backplate. The mount base is connected to the wing by the backplate, while the filter holder snaps into the base and secures the position of the filter holder.

[Figure]

Figure 2: 3D printed parts for the filter holder mounting to the plane. From left to right: mount base, mount backplate, filter holder mount and final setup. The mount base is connected to the wing by the backplate, while the filter holder snaps into the base and secures the position of the filter holder.
* * *
We agree that this sentence was not clear. Under start-up procedure we also included the time needed for the UAV to reach the targeted sampling altitude. We have changed this sentence therefore:

old The start-up procedure typically takes less than 10 minutes, depending on the targeted sampling altitude.

new The start-up procedure, including the ascend time, typically takes less than 10 minutes, depending on the targeted sampling altitude.
* * *
L102-103: So the pressure data is from which sensor (SHT40 or BME280)?

The pressure data is taken from the BME280, by averaging over the flight duration. We have changed the sentence to accurately reflect this:

old The pressure data is important for the setup since it is used to calculate the sampling flow.

new The pressure data from the BME280 is important for the setup since it is used to calculate the sampling flow.
* * *
L110: "until analyzed" – define how long in L110.

The time between filter sampling and analysis with INSEKT does depend on some factors, i.e. shipping and availability of the instrument. We understand that this comment asks in regard to the results shown in this paper. The data sets available at `https://radar.kit.edu/radar/en/dataset/ecljSTKjCuIoqEkr?token=sSJKlzwZKHYlpepdBzaK` contain the complete list of all samples shown with the start and stop time of the experiment (at INSEKT) as well as the start and stop time of the filter sample. The frozen fractions of the aerosol suspensions shown in Figures 6 and 7 were analysed with INSEKT between 2021-06-01 and 2021-06-09, while the filters were taken between 2021-04-19 and 2021-04-21. The filters were therefore stored before analysis for around 45 days. We have added a sentence in the section "Filter handling and subsequent offline INP analysis" to highlight this:

old

new The results shown in this paper are obtained from filters stored for around 45 days at $-20\,°C$ before analysis.
* * *
L120-130: This part can be much shorter by citing Schneider et al. INSEKT is a developed technique and does not need this much description in the current manuscript.

The configuration of INSEKT described by Schneider et al. 2021 is very similar to the described configuration in this paper, but there are some smaller differences. We have found the need to specify the used equipment in more detail to increase repeatability of the results after the published results from Barry et al. 2021 (e.g., differences between PCR plates). In addition, a cooling rate of $0.33\,K/min$ was used for the analysis in this paper compared to $0.25\,K/min$ in Schneider et al. 2021.
* * *
L147: Why 500L? A list of sampling conditions for all filters presented in this paper (starting – ending time, sampled air volume, and used suspension water volume at least) should be included in Appendix or SI.

We have added a table with this information to the Appendix. The 500 l was just a typical value to demonstrate the principle of the freezing assay and its lower detection limit, which is based on the amount of air sampled.
* * *
L168: ... feasibility, the frozen fraction of aerosol particles collected on the UAV and ...

We have changed the sentence:

old To demonstrate the scientific feasibility, the inverse of the liquid fraction, the frozen fraction, of the UAV and ground filters are compared to their respective handling blank filters taken during campaign 2.

new To demonstrate the scientific feasibility, the frozen fraction of the UAV and ground filters are compared to their respective handling blank filters taken during campaign 2.
* * *
Figures 5 & 6 & 7: AMT submission guide (Figs & Tables) states that labels of panels must be included with brackets around letters by lowercase. Also, state what vertical errors represent in the caption of Fig. 6 & 7.

We have updated the figures and added labels for the different panels. For Figures 6 and 7 we also changed the figure caption.
For Figure 6:

old The frozen fraction as a function of the freezing temperature $T$ is shown for the UAV filter suspensions from campaign 2 in comparison to its blank filter suspension and the Nanopure water background (left panel). The right panel shows the equivalent for the ground filters. The blanks were handled the same way as the filter, but the pump was not turned on, and the UAV was not flying (see Sect. 2.4).

new The frozen fraction as a function of the freezing temperature $T$ is shown for the UAV filter suspensions from campaign 2 in comparison to its blank filter suspension and the Nanopure water background (panel (a)). Panel (b) shows the equivalent for the ground filters. The blanks were handled the same way as the filter, but the pump was not turned on, and the UAV was not flying (see Sect. 2.4). The errorbars represent the 95 % confidence interval.

For Figure 7:

old The left panel shows the INP concentration in air, $c_{\mathrm{INP}}^{\mathrm{air}}$, at 400 m agl as a function of the freezing temperature, $T$, for a UAV and a ground (GR) filter during campaign 2. Both filters agree very well with each other. On the right panel, the same is shown for two filters one day after at 500 m agl. This filter was flown two times, doubling its sampling time and therefore increasing the amount of air sampled (note also the decreased lower detection limit as a red horizontal line, Eq. (3)). It can be seen that lower INP concentrations can be detected, *as well as a steeper freezing curve.* The freezing curve does not reach the water background on the right panel. This is due to the fact that no dilution was prepared, and therefore the water background was not reached with the higher amount of INPs that can freeze a well.

new Panel (a) shows the INP concentration in air, $c_{\mathrm{INP}}^{\mathrm{air}}$, at 400 m agl as a function of the freezing temperature, $T$, for a UAV and a ground (GR) filter during campaign 2. Both filters agree very well with each other. On panel (b), the same is shown for two filters one day after at 500 m agl. This filter was flown two times, doubling its sampling time and therefore increasing the amount of air sampled (note also the decreased lower detection limit as a red horizontal line, Eq. (3)). It can be seen that lower INP concentrations can be detected *due to the increased sampling time.* The freezing curve does not reach the water background on panel (b). This is due to the fact that no dilution was prepared, and therefore the water background was not reached with the higher amount of INPs that can freeze a well. The errorbars represent the 95 % confidence interval.

Note that the part in *italic* was changed due to comments from reviewer 1.
* * *
L187-188: This does not fit here. All of a sudden, the Arctic sampling with many citations kicks in here. Remote locations are not limited to the Arctic. Maybe it fits better as an outlook.

We have moved this sentence to the outlook.
* * *
Fig. 7 @ 400 m agl, why UAV INP conc. is higher than the ground-level INP conc.?

We are unable to give a definitive answer to this question. For this paper we focused on the scientific feasibility of the presented setup. More flights over a longer time period are needed to obtain a statistic on the actual difference between ground-based and UAV-based INP concentrations. As mentioned above we conducted a follow-up study during autumn 2022 (Pallas Cloud Experiment 2022) and the results of this study will be published in a special issue from Earth System Science Data.

We have adjust the wording:

old  Future *improvements* will include size distribution measurements in addition to INP measurements via small, lightweight optical particle counters.

new  Future *experiments* will include aerosol particle size distribution measurements in addition to INP measurements via small, lightweight optical particle counters.

Note that the part in *italic* was changed due to comments from reviewer 1.

**References**

Bärfuss, K. et al. (2018). "New Setup of the UAS ALADINA for Measuring Boundary Layer Properties, Atmospheric Particles and Solar Radiation". In: *Atmosphere* 9.1. ISSN: 2073-4433. DOI: 10.3390/atmos9010028.

Barry, K. R. et al. (2021). "Pragmatic protocols for working cleanly when measuring ice nucleating particles". In: *Atmospheric Research* 250, p. 105419. DOI: 10.1016/j.atmosres.2020.105419.

KNF (n.d.). *Micro Membran Gasförderpumpen.* BA321648-321650. KNF.

Lampert, A. et al. (2020). "Unmanned Aerial Systems for Investigating the Polar Atmospheric Boundary Layer—Technical Challenges and Examples of Applications". In: *Atmosphere* 11.4. ISSN: 2073-4433. DOI: 10.3390/atmos11040416.

Marinou, E. et al. (2019). "Retrieval of ice-nucleating particle concentrations from lidar observations and comparison with UAV in situ measurements". In: *Atmos. Chem. Phys.* 19.17, pp. 11315–11342. ISSN: 1680-7324. DOI: 10.5194/acp-19-11315-2019.

Schneider, J. et al. (2021). "The seasonal cycle of ice-nucleating particles linked to the abundance of biogenic aerosol in boreal forests". In: *Atmospheric Chemistry and Physics* 21.5, pp. 3899–3918. ISSN: 1680-7324. DOI: 10.5194/acp-21-3899-2021.

Villa, T., F. Gonzalez, B. Miljievic, Z. Ristovski, and L. Morawska (2016). "An Overview of Small Unmanned Aerial Vehicles for Air Quality Measurements: Present Applications and Future Prospectives". In: *Sensors* 16.7, p. 1072. DOI: 10.3390/s16071072.

Yu, F. et al. (2017). "Design and implementation of atmospheric multi-parameter sensor for UAV-based aerosol distribution detection". In: *Sensor Review* 37.2, pp. 196–210. DOI: 10.1108/sr-09-2016-0199.

---

## Author Response (AR2)

**Author response to RC2 - 2nd round**

February 25, 2025

We are grateful for the helpful comments and suggestions from the reviewer. The reviewers comments are in blue with our responses in black directly below. We have marked removed parts in red and added parts in green.

2nd round of review of "A novel aerosol filter sampler for measuring the vertical distribution of ice-nucleating particles via fixed-wing uncrewed aerial vehicles" by A. Böhmländer et al., submitted to AMT:

The manuscript improved somewhat, but there is still one open issue that needs to be addressed (1) and a few editorial comments (2). The editorial comments are minor and I try to improve readability by giving suggestions. But the one major issue is one I mentioned before, and which was not resolved sufficiently. Not much work is needed to resolve it, but it needs to be changed before I can recommend publication.

(1) One major issue Chapter 3.1 is called "Campaign 1". In your review you state that all you have to say about Campaign 1 is, that this was only a feasibility study and that no data is shown. From how I understand that now, and also referring to the figure caption of Fig. 6, both plots, Fig. 6 and Fig. 7, show INP data from Campaign 2. As such, these should both be discussed in the Chapter 3.2 called "Campaign 2". Right now, Fig. 6 (with data from Campaign 2) is introduced in the Chapter called "Campaign 1". And that is not only very confusing, but also gives a wrong impression. It also means that the modifications you discuss in the beginning of Chapter 3.2 were already in place for the data you show in Fig. 6. Therefore, also the sequence in your text is wrong. You need to change your chapter names and content, so that the chapter titles reflect the content. And you need to state more clearly, that no data from Campaign 1 is shown at all!

We agree that this was not clearly described by us. We have combined the two chapers "Campaign 1" and "Campaign 2" into a single chapter, called "Campaign 1 & Campaign 2", accordingly (Line 169).

old Campaign 1

new Campaign 1 & Campaign 2

We also adjusted the structure of the text to first give the changes on the setup, before providing the results obtained during campaign 2. We also added a sentence to clarify, that we do not show data from campaign 1 (Lines 170-195).

old Campaign 1 demonstrated the technical feasibility of the new UAV aerosol sampler in combination with the INSEKT INP analysis. To demonstrate the scientific feasibility, the frozen fraction of the UAV and ground filters are compared to their respective handling blank filters taken during campaign 2. Only one handling blank filter was taken for each setup. Figure 6 shows the frozen fraction as a function of the freezing temperature for all UAV filters, one blank filter, and its respective water background on panel (a). Panel (b) shows the same for the ground-based filters.

new Campaign 1 demonstrated the technical feasibility of the new UAV aerosol sampler in combination with the INSEKT INP analysis. No data obtained during campaign 1 is shown here. To demonstrate the scientific feasibility, the frozen fraction of the UAV and ground filters are compared to their respective handling blank filters taken during campaign 2*, when one handling blank filter was taken for the UAV and one for the ground. For campaign 2, the setup was modified slightly (see Sect. 2.2), and in addition, the same filter was used for sampling during two consecutive flights. The flow over the filter is calculated by the mean pressures during sampling. The actual flow is the weighted arithmetic mean, where the weight is defined by the sampling time for each flight.* Figure 6 shows the frozen fraction as a function of the freezing temperature for all UAV filters, one blank filter, and its respective water background on panel (a). Panel (b) shows the same for the ground-based filters.

Please note that the part in *italic* was changed due to one of the editorial comments (see below).
* * *
**Editorial comments**

Line 117: "(see e.g., Barry et al., 2021)." Should be ", see e.g., Barry et al., (2021)."

We have changed the citation to match this comment.

old  For a more detailed description of potential contaminations and procedures during filter handling
`\citep[see e.g.,][]{Barry2021}`

new  For a more detailed description of potential contaminations and procedures during filter handling
`, see e.g., \citet{Barry2021}`

resulting in

old  For a more detailed description of potential contaminations and procedures during filter handling (see e.g., Barry et al. 2021).

new  For a more detailed description of potential contaminations and procedures during filter handling, see e.g., Barry et al. 2021.
* * *
Lines 148-151: It is still weird that you first give a number for an estimation that you then reject, while afterwards describe an improved estimation but not give a numeric value for that better estimation. I suggest to at least rephrase everything from "This detection …" to "… earlier estimate" with something like: "This rough detection limit is, however, is still dependent on the fraction of the whole suspension used for an analysis. When accounting for the analysed water fraction, an improved detection limit results from:"

We have rephrased the wording and hope that it is now more clear. We also added a sentence to provide an example.

old  This detection limit also depends on the analysis, since only a fraction of the whole suspension is used for one analysis. Therefore, a better estimate of the lower detection limit is given by the product of the analysed water fraction and the earlier estimate

$$c_{\mathrm{INP,low}} = c^*_{\mathrm{INP,low}} \frac{V_{\mathrm{sol}}}{V_{\mathrm{well}} n_{\mathrm{filled}}} \; , \tag{3}$$

where $n_{\mathrm{filled}}$ is the number of wells filled with the suspension.

new  This rough detection limit however, does not factor in that only a fraction of the whole suspension is used for one analysis. When accounting for the analysed water fraction, an improved estimate of the lower detection limit is given by the product of the analysed water fraction and the earlier estimate

$$c_{\mathrm{INP,low}} = c^*_{\mathrm{INP,low}} \frac{V_{\mathrm{sol}}}{V_{\mathrm{well}} n_{\mathrm{filled}}} \; , \tag{3}$$

where $n_{\mathrm{filled}}$ is the number of wells filled with the suspension. For a typical solution volume of 5 ml and 64 wells used for the analysis, the improved estimate is around 1.5 higher than the rough estimate.
* * *
Line 172: The new sentence somewhat destroys the flow of the text (i.e., "Only one handling blank filter was taken for each setup"). I suggest to instead do not insert this new sentence, but replace this new sentence with: ", when one handling blank filter was taken for the UAV and one for the ground."

We have changed the sentence accordingly.

old  To demonstrate the scientific feasibility, the frozen fraction of the UAV and ground filters are compared to their respective handling blank filters taken during campaign 2. Only one handling blank filter was taken for each setup.

new  To demonstrate the scientific feasibility, the frozen fraction of the UAV and ground filters are compared to their respective handling blank filters taken during campaign 2, when one handling blank filter was taken for the UAV and one for the ground.
* * *
Line 204: "by increasing the sampled air" should better be "by increasing the volume of sampled air."

We have changed the sentence.

old  The INP detection limit can further be decreased by flying the same filter multiple times, therefore increasing the sampled air.

new  The INP detection limit can further be decreased by flying the same filter multiple times, therefore increasing the volume of sampled air.
* * *
Lines 218-219: "…, divided by the number of flights per filter, and dependent on the flight altitude." Both of these statements are confusing in the way how they are presented: The number you give in the first part of the sentence is for one flight. This detection limit you give is not yet "divided by the number of flights", but has to be if several flights are collected onto one filter. It would be good to start a new sentence on that topic where the citation above starts. And for "typical detection limits … around …", the flight altitude may not merit to be explicitly mentioned - its mention was more confusing than helpful.

We have removed the second part refering to the dependence on the altitude and started a new sentence to make it clear that the detection limit given can be improved by using the same filter on multiple flights.

old  Typical detection limits for the setup for one flight are around $c_{\mathrm{INP,low}}^{*} = 1 \times 10^{-3} \, l_{\mathrm{std}}^{-1}$, divided by the number of flights per filter, and dependent on the flight altitude.

new  Typical detection limits for the setup for one flight are around $c_{\mathrm{INP,low}}^{*} = 1 \times 10^{-3} \, l_{\mathrm{std}}^{-1}$. This detection limit can be decreased by sampling the same filter on multiple flights.
* * *
**References**

Barry, K. R. et al. (2021). "Pragmatic protocols for working cleanly when measuring ice nucleating particles". In: *Atmospheric Research* 250, p. 105419. DOI: 10.1016/j.atmosres.2020.105419.

---

## Author Response (AR3)

**Author response to EC1**

April 17, 2025

We are grateful for the helpful comments and suggestions from the editor. Below the editors comments are in blue with our responses in black directly below.

**1 General comments**

The authors present an insightful and valuable study on a critically needed vertical measurement of INPs. While they have made substantial improvements in their revisions, I have a few minor suggestions for consideration before publication. Although I have marked this as a "major revision," the suggested changes are relatively minor.
* * *
Line 35: The authors should include one of the earliest vertically resolved INP measurements via balloon by Creamean et al. (2018). I recommend adding this reference and briefly describing it, as done for Porter et al. (2020).

We thank the editor for this comment and have accordingly added the reference to Creamean et al. 2018 in line 35:

old Some studies have been performed to measure INPs on a UAV (Bieber et al. 2020; Schrod et al. 2017) or with balloon-based sampling systems (Porter et al. 2020).

new Some studies have been performed to measure INPs on a UAV (Bieber et al. 2020; Schrod et al. 2017) or with balloon-based sampling systems (Creamean et al. 2018; Porter et al. 2020).

as well as a short description of the measurements done by Creamean et al. 2018 in line 39:

old

new Creamean et al. 2018 developed a lightweight system to measure the INP concentration of aerosol particles deposited on a filter via a launched balloon. The system was tested up to an altitude of 1.1 km agl and also measures the total particle concentration.
* * *
Line 142: Creamean et al. (2024) should also be cited, as this updated DOE ARM report on the Ice Spectrometer provides additional details and images beyond Hill et al. (2016).

We appreciate this comment and have added the additional reference.

old For a more detailed look into INSEKT and the used formulas, see Hill et al. 2016; Schneider et al. 2021; Vali 1971.

new For a more detailed look into INSEKT and the used formulas, see Creamean et al. 2024; Hill et al. 2016; Schneider et al. 2021; Vali 1971.
* * *
Line 174: The data from campaign 1 should either be shown, or, the absence of data should be explicitly addressed. Figure 2 from the first review response contains useful data that should be incorporated into the manuscript as a proof of concept. This figure provides valuable supporting evidence. However, if the authors feel strongly about

[Figure]

Figure 1: The frozen fraction as a function of the freezing temperature $T$ is shown for a UAV filter suspension from campaign 1 in comparison to its Nanopure water background (panel (a)). Panel (b) shows the equivalent for the ground filters. The errorbars represent the 95 % confidence interval.

not including that plot, they should clarify why the data from Campaign 1 are not included, as their omission currently raises questions about its necessity to the study.

We have incorporated the figure into the main text of the manuscript. We updated the plot to fit to the style of the other figures and changed the following part to refer to it and describe it briefly.

old  No data obtained during campaign 1 is shown here.

new  The frozen fraction of a UAV and a corresponding ground filter from campaign 1 is shown in Fig. 1. The frozen fraction of the undiluted sample shows a clear separation from the water background.
* * *
Figures 6 and 7: The panels would be more visually accessible if they were wider rather than taller. I suggest adjusting their scale accordingly. Additionally, the label should read "frozen fraction" without the "/-" notation.

We have removed the notation on the y-axis label. We agree that a different layout will improve the visualization and have therefore switched the layout of the two graphs to allow for more spacing on the temperature axis. The old and new figure 6 are shown in figure 6 (old) and figure 6 (new), respectively, whereas the same is shown in figure 7 (old) and figure 7 (new) for figure 7. Please also note that the figure number has changed due to the addition of a new figure (see previous comment and figure 1).
* * *
Section 3.2: I recommend including a figure for campaign 3 similar to Figure 7 to further emphasize the importance and utility of vertical INP measurements. It is not clear why these data are not shown.

Campaign 3 was a test of the newest setup, but unfortunately due to bad weather and a short duration of the campaign, we were only able to collect three filters. Due to the low amount of samples, we would like to refer

[Figure]

Figure 6 (old): Old figure 6.

[Figure]

Figure 6 (new): New figure 6.

[Figure]

Figure 7 (old): Old figure 7.

[Figure]

Figure 7 (new): New figure 7.

to the extensive Pallas Cloud Experiment 2022 (PaCE-2022), where we collected aerosol filter samples using the setup described in this paper. A preprint describing the dataset obtained is available as a preprint in a special issue of Earth System Science Data (Böhmländer et al. 2025).
* * *
The authors note in their first review response that no significant difference was observed in INPs, suggesting that the sampling setup did not yield an improvement. Can they provide an explanation for this? A discussion of potential reasons would be a valuable addition to the manuscript.

We assume that his comment is in relation to our first response to reviewer 2, where we write: "Unfortunately, we are unable to directly report an improvement in the quality of the INP conentration measurements from the setup improvements." The setup definitely improved on the ease of handling as well as on the collection efficiency of smaller and larger aerosol particles. Since we do not know the size range of the ambient INPs, we are unable to quantify the improvement of the obtained INP concentration. It is true however, that the obtained INP concentration is more representative of the total aerosol population.
* * *
**2  Additional changes**

We identified two spelling mistakes, that we have fixed now:
in line 49:

old  duations

new  durations

and in line 166:

old  occurence

new  occurrence

**References**

Bieber, P. et al. (2020). "A Drone-Based Bioaerosol Sampling System to Monitor Ice Nucleation Particles in the Lower Atmosphere". In: *Remote Sensing* 12.3, p. 552. DOI: 10.3390/rs12030552.

Böhmländer, A. et al. (2025). "Measurement of the ice-nucleating particle concentration using a mobile filter-based sampler on-board of a fixed-wing uncrewed aerial vehicle during the Pallas Cloud Experiment 2022 [dataset]". In: *Earth System Science Data.* in preparation. DOI: 10.5194/essd-2025-87.

Creamean, J., T. Hill, C. Hume, and T. Devadoss (2024). *Ice Nucleation Spectrometer (INS) Instrument Handbook.* Tech. rep. DOE/SC-ARM-TR-278. Richland, Washington: U.S. Department of Energy, Atmospheric Radiation Measurement user facility.

Creamean, J. M. et al. (2018). "HOVERCAT: a novel aerial system for evaluation of aerosol–cloud interactions". In: *Atmospheric Measurement Techniques* 11.7, pp. 3969–3985. ISSN: 1867-8548. DOI: 10.5194/amt-11-3969-2018.

Hill, T. C. J. et al. (2016). "Sources of organic ice nucleating particles in soils". In: *Atmospheric Chemistry and Physics* 16.11, pp. 7195–7211. DOI: 10.5194/acp-16-7195-2016.

Porter, G. C. E. et al. (2020). "Resolving the size of ice-nucleating particles with a balloon deployable aerosol sampler: the SHARK". In: *Atmospheric Measurement Techniques* 13.6, pp. 2905–2921. DOI: 10.5194/amt-13-2905-2020.

Schneider, J. et al. (2021). "The seasonal cycle of ice-nucleating particles linked to the abundance of biogenic aerosol in boreal forests". In: *Atmospheric Chemistry and Physics* 21.5, pp. 3899–3918. ISSN: 1680-7324. DOI: 10.5194/acp-21-3899-2021.

Schrod, J. et al. (2017). "Ice nucleating particles over the Eastern Mediterranean measured by unmanned aircraft systems". In: *Atmos. Chem. Phys.* 17.7, pp. 4817–4835. DOI: 10.5194/acp-17-4817-2017.

Vali, G. (1971). "Quantitative Evaluation of Experimental Results an the Heterogeneous Freezing Nucleation of Supercooled Liquids". In: *J. Atmos. Sci.* 28.3, pp. 402–409. DOI: 10.1175/1520-0469(1971)028<0402:qeoera>2.0.co;2.